# Time-restricted feeding restores muscle function in *Drosophila* models of obesity and circadian-rhythm disruption

Jesús E. Villanueva[1,4], Christopher Livelo[1,4], Adriana S. Trujillo[1], Sahaana Chandran[1], Brendon Woodworth [1], Leo Andrade[2], Hiep D. Le[3], Uri Manor [2], Satchidananda Panda[3] & Girish C. Melkani [1,3]

Pathological obesity can result from genetic predisposition, obesogenic diet, and circadian rhythm disruption. Obesity compromises function of muscle, which accounts for a majority of body mass. Behavioral intervention that can counteract obesity arising from genetic, diet or circadian disruption and can improve muscle function holds untapped potential to combat the obesity epidemic. Here we show that *Drosophila melanogaster* (fruit fly) subject to obesogenic challenges exhibits metabolic disease phenotypes in skeletal muscle; sarcomere disorganization, mitochondrial deformation, upregulation of Phospho-AKT level, aberrant intramuscular lipid infiltration, and insulin resistance. Imposing time-restricted feeding (TRF) paradigm in which flies were fed for 12 h during the day counteracts obesity-induced dysmetabolism and improves muscle performance by suppressing intramuscular fat deposits, Phospho-AKT level, mitochondrial aberrations, and markers of insulin resistance. Importantly, TRF was effective even in an irregular lighting schedule mimicking shiftwork. Hence, TRF is an effective dietary intervention for combating metabolic dysfunction arising from multiple causes.

[1] Department of Biology, Molecular Biology Institute and Heart Institute, San Diego State University, San Diego, CA 92182, USA. [2] Waitt Advanced Biophotonics Center, Salk Institute for Biological Studies, La Jolla, CA 92037, USA. [3] Regulatory Biology Laboratory, Salk Institute for Biological Studies, La Jolla, CA 92037, USA. [4] These authors contributed equally: Jesús E. Villanueva, Christopher Livelo. Correspondence and requests for materials should be addressed to G.C.M. (email: gmelkani@sdsu.edu)

The prevalence of obesity and its associated comorbidities present one of the greatest challenges in modern health-care[1]. Pathological obesity is multifactorial and can result from genetic predisposition, a calorie-dense diet, and circadian disruption. Features of pathological obesity include the formation of ectopic lipids in non-adipose tissues and the disruption of glucose metabolism[2,3]. Skeletal muscle accounts for 40–50% of the total body mass and consumes 75–95% of insulin-mediated glucose, makes it the largest glucose-metabolizing organ[4]. Although muscle is not a primary fat-storing tissue, in obese patients excess fatty acids can be deposited in skeletal muscle cells as intramyocellular lipids or intramyocellular triglycerides (IMCLs or IMTGs, respectively). IMTGs are key components of IMCLs and are benign if regularly depleted, as observed in ath-letes[5]. However, the accumulation of IMTGs seen in sedentary, overweight individuals, can contribute to insulin resistance and tissue dysfunction[6].

Most animal models of obesity fall into two categories: diet-induced obesity (DIO), or genetic-induced obesity (GIO). Diets that include excess fat and/or sugar cause obesity and are asso-ciated comorbidities in humans, rodents, and flies [7–10]. Drosophila have been used to model human metabolic diseases due to con-served mechanisms of nutritional sensing, energy utilization, and energy storage[9,11,12]. Relevant to this study, the Drosophila mus-cular system has been used to define the molecular basis for muscle organization, aging, and disease[12,13]. Importantly, the genetic manipulation of muscle cells in rodents often results in lethality, but the same manipulations in Drosophila indirect flight muscles (IFMs) do not[13–16]. Thus, progressive muscle degeneration can be studied in Drosophila in the absence of complicating systemic effects.

Drosophila that lack sphingosine kinase 2 (Sk2) have served as a model of genetic obesity and lipotoxic cardiomyopathy[17,18]. Similarly, loss-of-function mutations with two additional genes have also used as models of genetic obesity. Brummer (Bmm) encodes triglyceride lipase, which is homologous to the mam-malian adipose triglyceride lipase (ATGL). Infertile crescent (Ifc) encodes sphingosine Δ−4 desaturase, which also has human homologue[17,19,20]. Despite the importance of the skeletal muscle as a metabolic tissue, its contribution to genetic obesity-related pathophysiology has been grossly overlooked.

Circadian-rhythm disruption (CRD) is emerging as another risk factor for obesity and metabolic diseases. Through direct and indirect mechanisms, circadian clock genes modulate the tem-poral expression and function of metabolic regulators, thereby optimizing nutrient metabolism in relation to daily cycles in food availability and energy expenditure[11,21–23]. Perturbation of cir-cadian rhythms results in dysmetabolism and obesity-associated comorbidities, such as diabetes and cardiovascular disease[24,25]. Although circadian oscillators are based on self-sustained and cell-autonomous transcriptional feedback loops, their synchrony and oscillation amplitudes rely on environmental signals, such as light–dark periodicity, time of feeding, and sleep[11,23]. Accord-ingly, constant light, aberrant eating schedules, and sleep dis-ruption all predispose individuals to obesity-associated dysfunction and exacerbate the resulting phenotypes.

Conversely, sustaining daily rhythms in feeding and fasting under time-restricted feeding (TRF) without reducing caloric intake promotes daily rhythms in gene expression and function, and can prevent and even reverse obesity and metabolic disorders[26,27]. Several benefits of TRF are described in rodents and Drosophila[28,29]. These observations suggest that interven-tions that improve molecular circadian rhythms are potential entry points for managing the risk of obesity and metabolic diseases[30]. Pilot studies are beginning to show TRF can reduce the risk of metabolic disease among overweight or obese humans[31,32].

While most mechanistic TRF studies have examined in the mouse liver or Drosophila heart, the impact of TRF on skeletal muscle function remains under investigated.

Here, we assess the effects of genetic obesity, HFD-induced obesity, and CRD on skeletal muscle function in Drosophila. All three obesogenic challenges compromised muscle physiology. We found that TRF improves muscle physiology in these models of obesity, alleviating phenotypes related to muscle dysfunction, sarcomere disorganization, mitochondrial abnormalities, lipid infiltration, and insulin resistance, without affecting lifespan. These analyses suggest that TRF can reduce the adverse metabolic effects of obesity and CRD, resulting in the attenuation of muscle dysfunction.

## Results

**Drosophila models of genetic and diet-induced obesity.** To model diet-induced obesity, we subjected 4-day-old wild-type Drosophila (WT; Canton-S and $w^{1118}$) to either a high-fat-diet (HFD, base yeast fly diet supplemented with 5% coconut oil), a high-glucose/sugar diet (HSD, base diet supplemented with 300 mM sugar), or a HFD + HSD combination (base diet supple-mented with 5% coconut oil and 300 mM sugar) (Fig. 1a). To model genetic predisposition to adiposity, we analyzed loss-of-function Sk2, Ifc, and Bmm mutants[17,19,20]. These mutants were fed with regular diet (RD) or HFD.

After 3 weeks of eating their assigned diet, both DIO and GIO models showed weight gains of up to 30% (Fig. 1b). To determine whether DIO or GIO resulted in increased ectopic fat storage, we measured TG levels in the IFMs of obese flies. Indeed, GIO flies displayed TG increases of ~55%, HSD and HFD resulted in increases of ~12% and ~60%, respectively (Fig. 1c). The HFD + HSD diet had a cumulative effect on TG, resulting in a TG increase of 87%, which was greater than the sum of the individual diets.

To test the effects of obesity-induced metabolic dysregulation on skeletal muscle performance and locomotion, we performed commonly used flight and climbing assays[14] (see below). DIO and GIO models exhibited declines in both climbing ability and flight performance (Fig. 1d, e). For DIO, HSD had the smallest effect on performance ~18%, in both assays, followed by HFD ~43%, and HFD + HSD ~68%. All models of GIO exhibited reductions between 13–44% compared with background-matched controls. Obese flies, when compared with age-matched controls, exhibited higher body weight, which mirrored a decline in skeletal muscle assays (Fig. 1b). However, within the obese group, weight was not directly correlated with muscle dysfunction (red asterisks in Fig. 1b, d, e).

Obesity is a precursor to insulin resistance and increased circulating sugar. We quantified hemolymph glucose and trehalose levels in flies fed HFD and in Sk2 mutants[7,33]. Both models of obesity showed elevated levels of glucose and trehalose in the hemolymph (Fig. 1f; Supplementary Fig. 1e). In addition, we measured RNA levels for Neural Lazarillo (NLaz; homologue of the human Apolipoprotein D), which is a marker for insulin resistance in flies fed HSD[34]. Both DIO and GIO models exhibited greater than twofold increase in NLaz expression in their IFMs compared with age-matched RD controls (Fig. 1g).

The cytotoxic effect of adipose dysfunction is mediated by the accumulation of ectopic fat in non-adipose tissues, such as skeletal muscle[35]. Cryosections of IFMs from 3-week-old female Canton-S flies under HFD and genetically obese Sk2 mutants under RD exhibited actin-containing myofibrillar disorganization (phalloidin staining) and the appearance of intramuscular ectopic fat (Nile red staining). Both DIO and GIO had a ≥ 3.8-fold larger

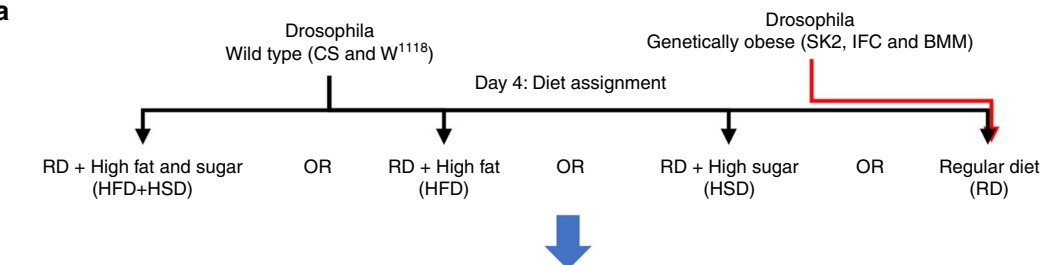

Nile Redpositive area representing ectopic lipid deposits, compared with RD flies (Fig. 1h, i).

Ectopic fat accumulation compromises mitochondrial function and promotes the production of reactive oxygen species[36]. We measured oxidative mitochondrial stress of skeletal muscle from 3-week-old flies using the MitoTimer reporter gene (Fig. 1j, k). The MitoTimer reporter expresses a mitochondria-specific protein that irreversibly changes fluorescence from green to red upon oxidation[37]. Red punctae were clearly visible in cryosectioned flight muscles of HFD and *Sk2* mutants (Fig. 1j), suggesting the induction of oxidative mitochondrial damage. Red-to-green ratios of female HFD and mutant *Sk2* flies increased by ≥ 2.4 fold (Fig. 1k).

High fat in particular contributed to a drastic reduction in lifespan ≥ 50%, Supplementary Fig. 2, Supplementary Tables 1, 2). Although the cytological and molecular signatures of compromised muscle correlated with reduced flight index and geotaxic performance, we acknowledge that other factors, including innervation, non-muscle function or neuronal function, may have contributed to HFD-induced muscle dysfunction. Measurement of group circadian activity[10] in 3-week-old flies revealed that both obesity models exhibited activity levels comparable with age-matched wild-type flies (Supplementary Fig. 1g, n = 50). Minor differences between day and nighttime activity were seen, but this difference was statistically nonsignificant. These results support the interpretation that the compromised muscle function

**Fig. 1** Diet and genetic obesity-induced skeletal muscle physiology, lipid dysregulation and insulin resistance. **a** Diet and genetic-induced obesity model flow chart. At 3 weeks of age flies were tested for obesity-induced phenotypes in regards to, skeletal muscle physiology, and performance. **b** Normalized total body weight of 3-week-old wild-type (WT) $w^{1118}$ and CS, and obese male flies. Flies tested per cohort $N = 30$ from three independent experiments. **c** Normalized triglyceride levels were quantified from 3-week-old male flight muscle (IFM; indirect flight muscle) extracts. $N = 3$ independent experiments from 30 flies. **d**, **e** Climbing and flight-performance indices of 3-week-old female flies reveal functional defects resulting from diet and genetic-induced obesity ($N = 87$–100 under each condition from 8–10 independent experiments). **f** Hemolymph glucose content was significantly elevated in diet and genetically obese flies in 3-week-old female compared with age-matched control. $N = 3$ independent experiments from extracted Hemolymph from 100–150 flies. **g** NLaz RNA was upregulated in the IFM of diet and genetically obese flies in 3-week-old females. $N = 3$ independent experiments from 30 flies IFM. **h** Confocal images of the IFM from 3-week-old females of wild-type (CS-WT), WT fed with HFD and genetically obese SK2 flies probed with phalloidin (green) and Nile Red (red). Actin-containing myofibrillar disorganization is shown with asterisks and enhanced lipid accumulation in the skeletal muscle is shown with arrows. Scale bar is 20 μm. **i** Intramuscular lipid area quantification. $N = 9$. **j** Confocal images of the IFM of 3-week-old females expressing MitoTimer, a mitochondrial reporter for oxidative stress. Compared with wild-type, DIO and GIO flies showed enhanced red fluorescence. Scale bar is 20 μm. **k** Red-to-Green MitoTimer ratio quantification. $N = 3$. All statistical analyses except panel **d** were carried out using one-way ANOVA with post hoc Tukey test and for panel **d**, post ad hoc Sidak's method with multiple comparisons used. Bar graphs (along with dot plots also shown, where $n < 10$) and error bars indicate as mean ± SD, where * $p < 0.05$; ** $p < 0.01$*** $p < 0.001$; ns non significant. Statistical comparisons performed between test conditions and their genetic controls counterparts unless otherwise indicated. Black asterisks in all panels represent statistical differences relative to control and red asterisks in panels in **b**, **d**, and **e** represent statistical difference between Bmm and Sk2 mutants

is not a secondary effect of a wider perturbation of neuronal function.

**Obesity-induced myofibril and mitochondrial abnormalities.** We used transmission electron microscopy (TEM) of IFM sections[13,14] to assess the muscle ultrastructure. Longitudinal sections of Canton-S IFMs from 3-week-old adults showed regular myofibrillar organization with intact Z-discs and M-lines. Mitochondrial (m) morphology was normal with intact cristae (Fig. 2a, low- and high-magnification images). In contrast, IFMs from age-matched HFD flies showed mitochondrial abnormalities, including significant gaps in the cristae and mitochondrial degeneration (Fig. 2b, arrows). Similar phenotypes were also observed in 3-week-old Sk2 mutants (Fig. 2c). Both obesity models also showed sarcomere disorganization, including sarcomere cracking, which was more severe in the GIO model (Fig. 2 d, e, white arrowheads). Recent studies in mammalian models of obesity have also revealed myofibril disorganization, fibrosis, and mitochondrial defects in both cardiomyocytes and skeletal myocytes[38,39]. Quantification of the TEM images revealed that obesity did not affect sarcomere length but did disrupt the M-lines and Z-disks and caused severe mitochondrial abnormalities (Supplementary Fig. 1f). Compared with age-matched controls; ~15% and ~30% of sarcomeres possessed non-intact M-lines and/or Z-disks in addition to overall myofibrillar disarray seen in HFD flies and Sk2 mutants, respectively. Mitochondrial abnormalities were even higher in both obesity models, as ~35% and ~50 % of the mitochondria were abnormal in HDF and Sk2 mutants, respectively, compared with age-matched WT control (Supplementary Fig. 1f).

**Genetic perturbation of Sk2 causes tissue autonomous muscle defects.** In Sk2 mutants, expression of Sk2 is lost from all cells, including the skeletal muscle. To test whether observed muscle phenotypes are tissue autonomous, we performed RNAi knock-down (KD) of Sk2 specifically in IFMs[14]. We used a recombinant stock[14] possessing Fln-Gal4 (the flightin gene promoter drives expression specifically in the IFMs) and UAS-mito-GFP, which labels the mitochondria with GFP[16]. IFM-specific KD of Sk2 resulted in significant and progressive muscle dysfunction (flight defects) in both gender (Fig. 3a, $n = 100$–150 per genotype at each age). Gal4 alone did not affect muscle function in young flies, and exhibited age-dependent decline in muscle dysfunction (Fig. 3a). This muscle dysfunction obtained with IFM-specific KD of Sk2 is consistent with the Sk2 mutant (see below).

As seen in Sk2 mutant flies, IFM-specific KD of Sk2 upregulated NLaz (marker of insulin resistance) in IFMs of 3-week-old male and female flies, compared with age-matched driver controls (Fig. 3a, b). To determine the cellular consequences of muscle-specific KD of Sk2, IFMs were cryosectioned and viewed by confocal microscopy as described[14]. As in Sk2 mutants (Fig. 1h), IFM-specific KD of Sk2 resulted in myofibrillar disorganization and accumulation of ectopic fat (red puncta) in 3-week-old female flies, compared to age-matched driver controls (Fig. 3c, d). Using the mito-GFP marker, we also evaluated the impact of Sk2 KD on mitochondrial morphology. IFM-specific KD of Sk2 resulted in myofibrillar disorganization along with the fragmented and deformed mitochondria, whereas these phenotypes were not observed for the Gal4 driver alone (Fig. 3e, f). Overall most of the physiological, cytological, and metabolic parameters observed with muscle-specific KD of Sk2 are comparable with those seen in Sk2 mutants, suggesting that Sk2 mutants are obese from a muscle-specific effect.

**ATGL-mediated lipid turnover and the regulation of muscle physiology.** We next sought to determine whether the major triglyceride lipase, ATGL, encoded by the Bmm gene in Drosophila, influences muscle function. Compared with age-matched controls, heterozygous Sk2 and Bmm mutants exhibited reduced muscle performance. Tests were performed at 3 weeks of age in male and female flies (Fig. 4a, b, $n = 100$ for each genotype). As expected, muscle performance in Sk2 homozygous mutants were worse than Sk2 heterozygotes (Fig. 4a, b). Interestingly, muscle performance was severely compromised (Fig. 4a, b) in transheterozygotes (SK2/+ ; Bmm/+ ), and this was even lower than Sk2 homozygous mutants (Fig. 4a, b). Similarly, DIO muscle dysfunction was further deteriorated in the presence of Bmm mutant (Fig. 4a, b). Compared with Bmm/+ or HFD-induced muscle function was significantly reduced when HFD was used on Bmm/+ background (Fig. 4a, b).

To directly address if reduced muscle performance of the transheterozygote mutant (Sk2/+ ; Bmm/+ ) affects infiltration of ectopic fat into the skeletal muscle, and causes more severe myofibrillar disorganization, we used a cytological approach. Individual Sk2 and Bmm mutants in heterozygous state showed some myofibrillar disorganization (asterisks) and ectopic fat accumulation (red puncta), compared with age-matched control (Fig. 4c, d). However, these cytological defects were less severe than those seen in age-matched homozygous Sk2 mutants (Fig. 4c, d). Transheterozygote mutant (Sk2/+ ; Bmm/+ ) exhibited significant upregulation of ectopic fat infiltration into the skeletal

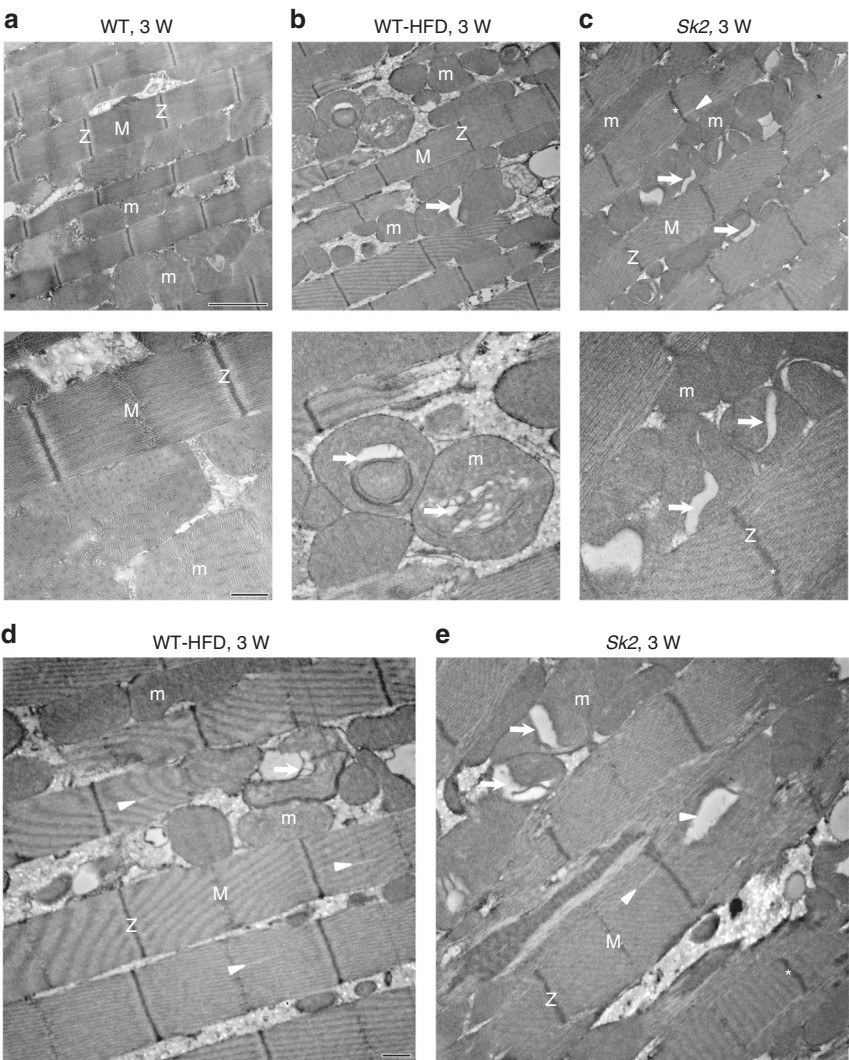

**Fig. 2** Obesity caused abnormal myofibrillar organization and mitochondrial defects. **a–e** Ultrastructural analysis using TEM revealed that obesity caused myofibril disorganization and mitochondrial abnormalities compared to wild-type CS under ALF in 3-week-old female flies IFM. Compared with control (**a**), occasional myofibrils with partially intact Z-disks (Z) and M-lines (M) were observed (**b**, **d**) with the diet-induced obesity, which were more prominent in the genetic obesity mutant (**c**, **e**). Abnormal mitochondria (m, arrows) and disorganized myofibrils (white arrowheads) were evident with diet-induced obesity. Genetic obesity also resulted in abnormal mitochondria (m, arrows), a more severe myofibril disorganization (white arrowheads) and resulted in wavy sarcomere and Z-disks (asterisks). No such abnormalities were seen in the age-matched control. Scale bar for the top **a–c** panels is 2 μm and bottom **a–c** panels is 0.5 μm. Scale bar for the panels **d** and **e** is 0.5 μm. The quantification of obesity-induced myofibrillar disorganization and mitochondrial abnormalities are shown in supplementary Fig. 1f

muscle and severe disorganization of myofibrils (Fig. 4c, d). Ectopic fat infiltration (arrows) and myofibrillar disorganization (asterisks) seen in transheterozygotes mutant were significantly higher than in each mutation (*Sk2 or Bmm*) in heterozygous state (Fig. 4c, d). Similarly, DIO lipid infiltration and myofibrillar disorganization was further enhanced when carried out in the *Bmm* mutant background (Fig. 4c, d). Compared with HFD or *Bmm/ +* alone, feeding *Bmm/ +* a HFD further elevated lipid accumulation and myofibrillar disorganization (Fig. 4c, d).

We also used a pharmacological approach to inhibit Bmm function in vivo using Bromoenol Lactone (BEL)[14]. Inhibition of ATGL with BEL resulted in physiological abnormalities which were more severe in the DIO and GIO compared with age-matched control (Supplementary Fig. 3). Compared with WT control, HFD and *Sk2*, feeding of 50 μM BEL leads to more lipid infiltration in the skeletal muscle (Supplementary Fig. 3a, b). Moreover, this additional lipid infiltration induced with BEL

feeding, resulted in more severe muscle dysfunction in the WT control and obesity background, compared with control (Supplementary Fig. 3c). Thus, both genetic and pharmacological inhibition of ATGL further deteriorated obesity-induced abnormalities.

Overall, using physiological, metabolic, and ultrastructural analyses (Figs 1–4), we found obesogenic challenges resulted in skeletal muscle dysfunction, myofibril disorganization, lipid infiltration, and mitochondrial stress.

**TRF restores muscle dysfunctions associated with age and obesity.** To test whether TRF can alleviate obesity-associated skeletal muscle dysfunction in age-, diet-, and genetic models of Drosophila, we subjected flies to either ad libitum feeding (ALF) (i.e., unrestricted access to food), or TRF, where flies were allowed to consume food only during the 12 h of daytime (Fig. 5a). Measurement of the daily net volume of food consumption

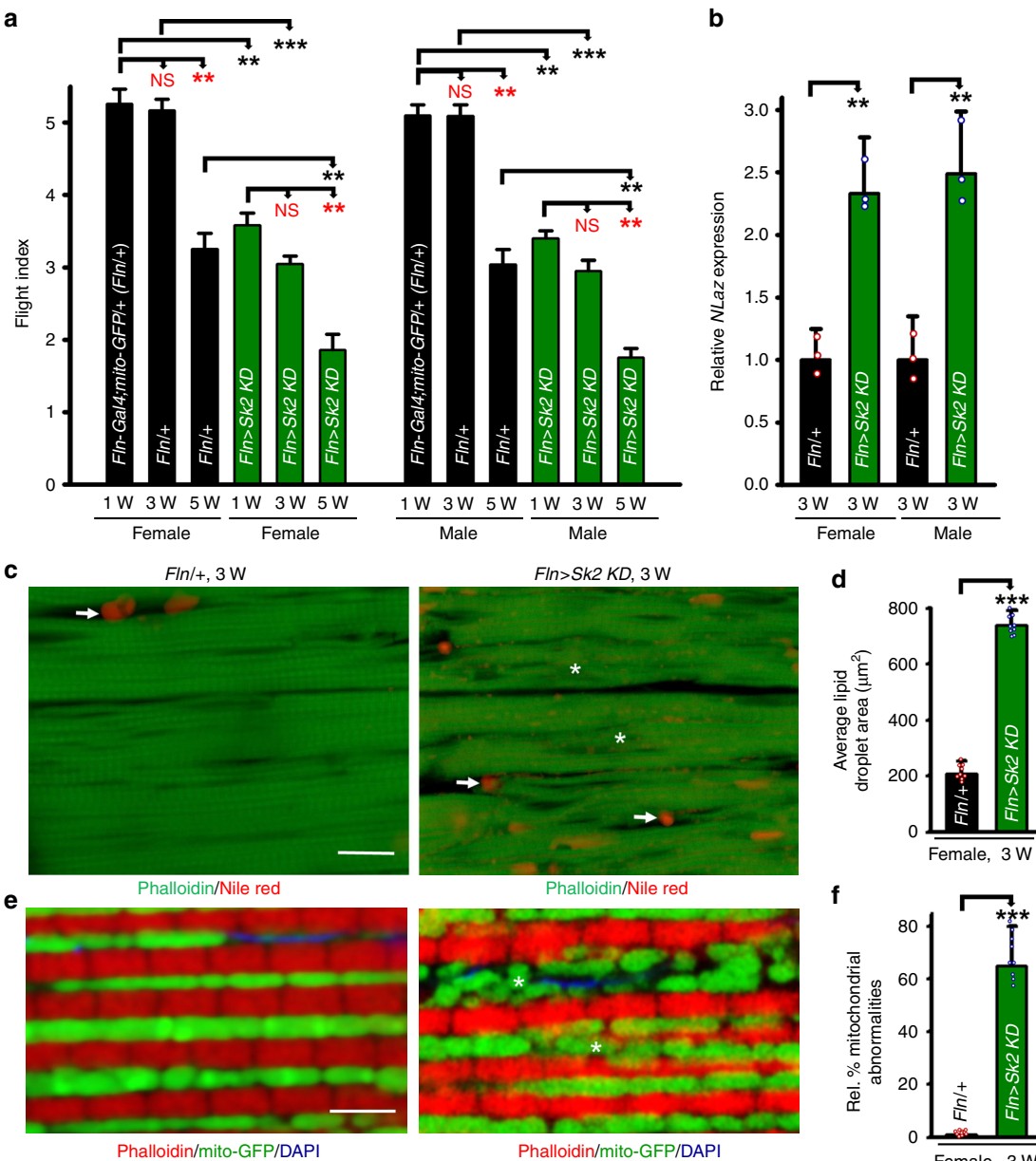

**Fig. 3** Muscle-specific *knockout* of *Sk2* resulted in physiological dysfunction, like the *Sk2* mutant. **a** Flight performance assay. Fly flight paths were recorded and given a numerical value to calculate flight index (F.I.) (see the Methods section). IFM-specific RNAi knock-down of *Sk2* using *Fln-Gal4;UAS-mito-GFP* (referred as Fln/ + ) resulted in a significant reduction in muscle performance in 1-, 3-, and 5-week-old male and female flies compared with age-matched driver control (*Fln/ + *). Age-dependent decline in flight performance is also shown in the *Sk2 KD* and driver control flies. (*n* = 100–150 flies were tested at each age in both male and female flies for the *Sk2 KD* and control flies from three independent experiments). **b** *NLaz* RNA was upregulated in the IFM upon *KD* of *Sk2* in the genetically obese flies (both male and female) after 3 weeks. *N* = 3 independent experiments from 30 flies IFM. **c** Confocal images of the IFM from 3-week-old females of wild-type (*Fln/ + *) and IFM-specific *KD* of genetically obese *Sk2* flies probed with phalloidin (green) and Nile Red (red puncta). More intramuscular lipid (arrows) and actin-containing myofibrillar disorganization was detected in IFM of *Sk2 KD* (asterisks), compared with age-matched driver control. Scale bar is 20 μm. **d** Intramuscular lipid area quantification showed significant increase upon *KD* of *Sk2* compare to age-matched driver control. *N* = 9 from three flies IFM in each genotype. **e** Confocal images of the IFM of 3-week-old females expressing mito-GFP (green), a reporter for mitochondrial morphology, DAPI (blue) and phalloidin (red). Compared with wild-type, *KD* of *Sk2* flies showed mitochondrial dysmorphology/abnormality (asterisks) and disorganization of actin-containing myofibrils. Scale bar is 4 μm. **f** Quantification of *Sk2* knockdown mitochondrial dysmorphology from three representative experiment shown. *N* = 9. For panels **a**, **b**, **d**, and **f** bar graphs (along with dot plots for panels **d** and **f**) and error bars presented as mean ± SD, and statistical analysis was performed using one-way ANOVA with post hoc Tukey test, where * $p < 0.05$, 0.01 $p < 0.01$, *** $p < 0.001$, ns nonsignificant

revealed that wild-type strains (Canton-S and w[1118]) and *Sk2* mutants (except w[1118] females) consumed equivalent amounts of food under ad libitum or TRF conditions (Supplementary Fig. 1a, b), and hence 12 -h TRF paradigm does not reduce caloric intake in flies.

We tested the effect of TRF on muscle function in young, middle-aged, and old flies using flight performance and locomotor assays (Fig. 5b, c, *n* = 100–250 strain/condition combination). In agreement with age-dependent functional declines[10,40], both RD males and females exhibited an age-

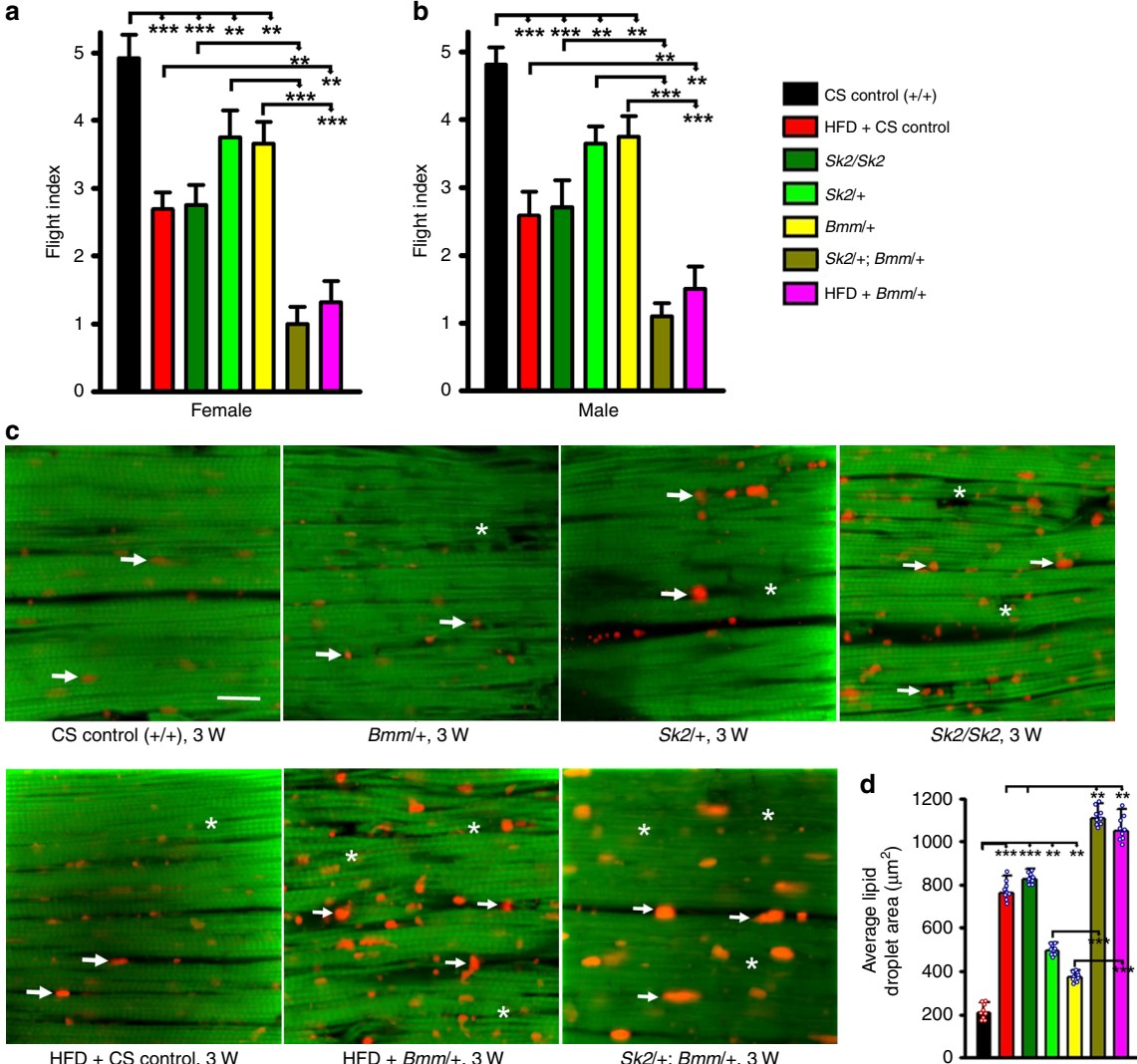

**Fig. 4** ATGL (*Bmm*)-mediated lipid turnover and regulation of obesity-induced muscle dysfunction. **a–b** Flight performance assay. Fly flight paths were recorded and given a numerical value to calculate flight index (F.I.) (see the Methods section). Muscle performance of each heterozygous mutant (*Bmm/+* and *Sk2/+*), homozygous mutant (*Sk2/Sk2*), transheterozygotes mutant (*Sk2/+; Bmm/+*), WT and Bmm/+ in the presence of HFD at the 3-week old in both male and female flies were significantly reduced compared with age-matched CS control (+/+). In addition, muscle performance of the transheterozygotes mutant (*Sk2/+; Bmm/+*) mutant and HFD + Bmm/+ were found significantly reduced compared to each age-match heterozygous mutant (*Bmm/+* and *Sk2/+*) or homozygous mutant (*Sk2/Sk2*) or HFD. (n = 100 flies were tested at each genotype in both male and female flies from three independent experiments). **c** Confocal images of the IFM from 3-week-old females of wild-type (+/+) each heterozygous mutant (*Bmm/+* and *Sk2/+*), homozygous mutant (*Sk2/Sk2*), and transheterozygotes mutant (*Sk2/+; Bmm/+*) mutant, HFD + WT and HFD + Bmm/+ probed with phalloidin (green) and Nile Red (red). More intramuscular lipid (arrows) and actin-containing myofibrillar disorganization was detected in transheterozygotes mutant (*Sk2/+; Bmm/+*) or HFD + Bmm/+ (asterisks), compared with any other mutants or HFD + WT. Scale bar is 20 μm. **d** Quantification of the intramuscular lipid area showed significant increase in each mutant compared to control and significantly elevated in transheterozygotes mutant (*Sk2/+; Bmm/+*) or HFD + Bmm/+, compared with age-matched control, other mutants or HFD + WT. N = 9 images from three different for each genotype. For panels **a**, **b**, and **d** bar graphs (along with dot plots for panel **d**) and error bars, presented as mean ± SD and statistical analysis was performed using one-way ANOVA with post hoc Tukey test, where * $p < 0.05$, 0.01 $p < 0.01$, *** $p < 0.001$, ns nonsignificant

associated decline in flight performance and climbing ability (Fig. 5d–i). Feeding the same flies, a HFD ad lib accelerated decreases in functional performance and locomotion. This trend continued throughout the experiment (Fig. 5d–i). Similarly, genetically obese *Sk2* mutants fed RD exhibited reduced flight and climbing performance compared with Canton-S controls (Fig. 5f, i for flight and geotaxis, respectively). TRF attenuated age-associated declines at most time points when animals were fed RD, and partially rescued performance in HFD flies (Fig. 5d–i, n = 100–250 in each

condition). TRF benefits were found for both male and female flies.

**TRF improves metabolic parameters in obesogenic challenged flies.** TRF prevents body weight gain in wild-type flies[10]. Here, we have shown that TRF flies-maintained baseline body weight in both diet and genetic models of obesity (Fig. 6a, n = 30, p < 0.05). As excess body weight is likely due to excess adiposity, part of which is stored in muscle, we tested the effect of TRF on muscle triglyceride content. TRF suppressed IFM triglyceride levels in

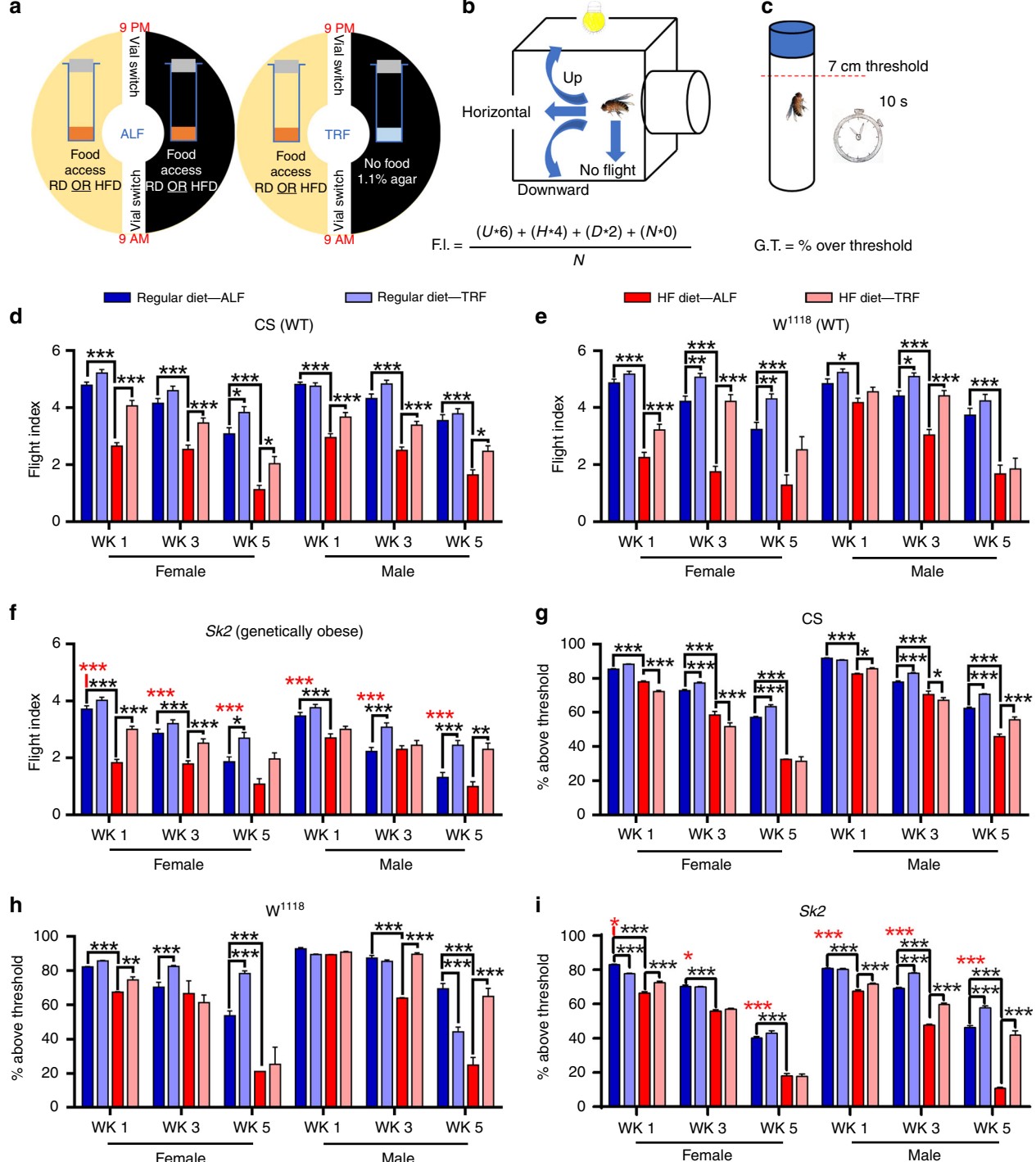

**Fig. 5** Time-restricted feeding attenuates age-related and obesity-induced dysfunction in the skeletal muscle. **a** Time-restricted feeding regimen. Wild-type and *Sk2* mutant flies were placed in either of two feeding regimens, ALF or TRF, and either RD or HFD (see Fig. 1a). ALF flies had unrestricted access to food throughout the day compared to TRF flies which were only allowed to eat during the first 12 h and were transferred to 1.1% agar during the nighttime. **b** Flight performance assay. Fly flight paths were recorded and given a numerical value to calculate flight index (F.I.) (see the Methods section). **c** Negative geotaxic assay. Fly vials were tapped gently, and flies allowed to climb for 10 s. The average percentage past a 7-in threshold was calculated (see Methods). **d–f** TRF suppresses obesity-induced skeletal muscle dysfunction. Flight performance for control and obese flies under ALF and TRF was carried out at 1, 3, and 5 weeks of age. Red asterisks indicate comparisons with CS RD ALF. Statistical comparisons are shown: ALF versus TRF and RD ALF versus HFD ALF. **g–i** TRF ameliorates declines in geotaxis performance. Red asterisks indicate comparisons with CS RD ALF with *Sk2* or HFD. Statistical comparisons: ALF versus TRF and RD ALF vs HFD ALF. The data presented as mean ± SD and statistical analysis was performed using ANOVA with post hoc Tukey test (panels **d**, **e**) and post ad hoc Sidak's method with multiple comparisons (panels **f–i**), from $N = 150–250$ (from 10–12 independent experiments) for each condition, where $* p < 0.05$; $** p < 0.01$; $*** p < 0.001$; ns nonsignificant

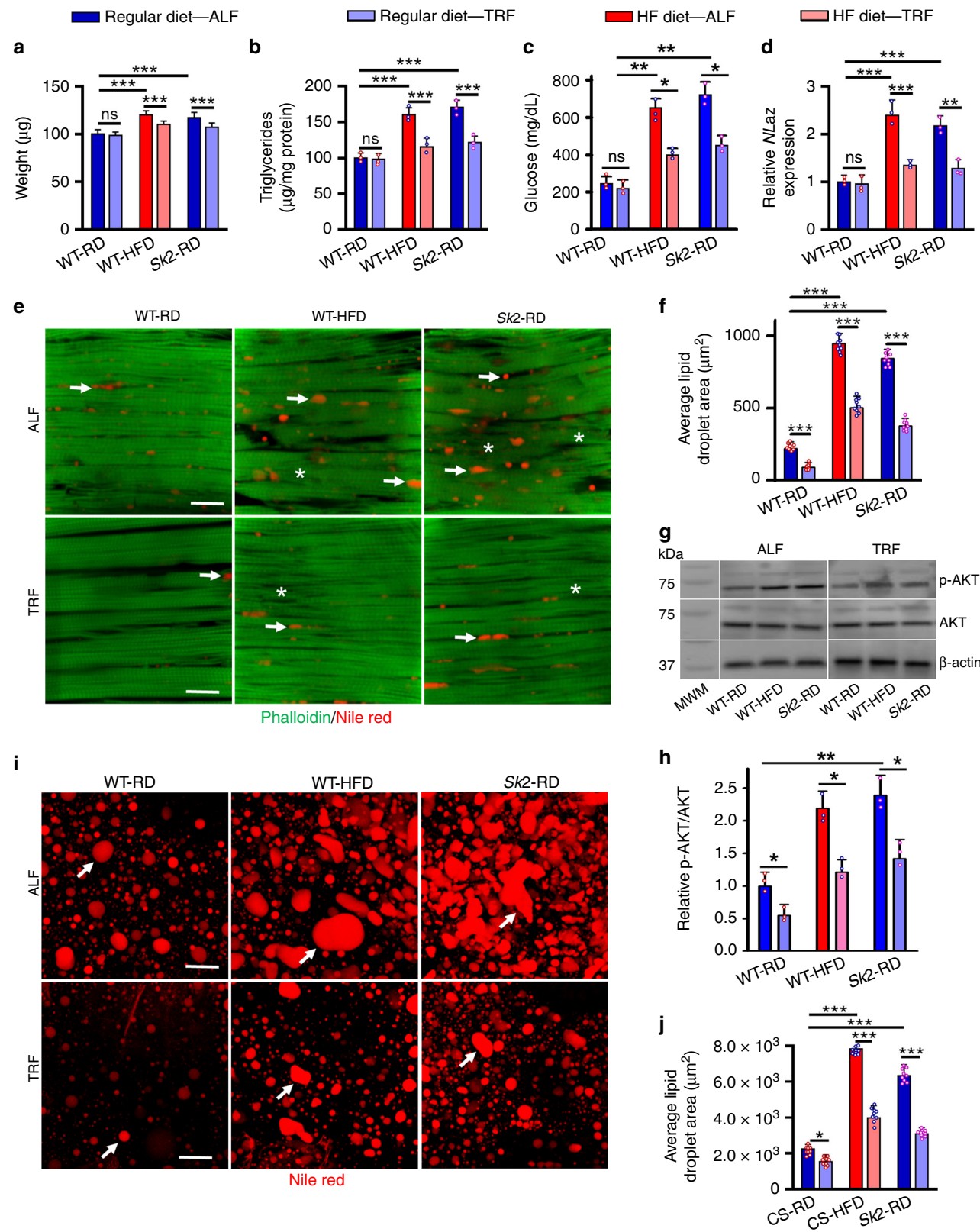

3-week-old males fed with a HFD, and *SK2* flies fed with a RD by ~21% ($n = 3$, $p < 0.001$), relative to those under ALF conditions. This reduced level was comparable with that in wild-type flies fed with a regular diet (Fig. 6b). Next, we tested if TRF was able to reduce levels of glucose and trehalose in the hemolymph of DIO and GIO models. In both models, levels of glucose and trehalose were reduced under TRF compared with RD (Fig. 6c; Supplementary Fig. 1e). To determine if the reduction in body weight, glucose, and triglyceride content correlated with enhanced insulin sensitivity, we measured levels of *Nlaz* expression. Interestingly,

**Fig. 6** TRF protects against obesity-associated weight gain, lipid dysregulation and insulin resistance. **a** Normalized body weight of 3-week-old male flies. **b** Normalized triglyceride content was measured from IFM extracts of 3-week-old male flies. ($N = 30$ from three independent experiments for **a**, **b**). **c** Hemolymph glucose content under ALF was found to be significant elevated in diet and genetically obese flies after 3 weeks compared to age-matched control female, which was significantly reduced under TRF in both obesity models, but not in the wild-type control. $N = 3$ independent experiments from extracted Hemolymph from 100–150 female flies per genotype. **d** TRF suppressed the upregulation of *NLaz* associated with obese models. *NLaz* RNA expression normalized to RPL11. **e** Confocal images and quantification show TRF suppresses obesity-induced lipid accumulation and myofibril disorganization in the skeletal muscle in 3-week-old WT-HFD flies and *Sk2*. Arrows and asterisks show the appearance of intramuscular lipids and myofibril disorganization respectively in ALF flies compared with TRF. Scale bar is 20 µm. Nile red quantification of lipids in the skeletal muscle (**f**) in 3-week-old female fly IFMs showed upregulation of fat contents in obese models and suppression under TRF. **g** Representative western blots of p-AKT protein levels (top), the total AKT (middle) normalized to β-actin, in IFMs from 3-week-old female flies WT, HFD, and *Sk2* mutant under ALF and TRF. MWM is referring to molecular weight markers. **h** Relative ratio of the p-AKT/total AKT represented to evaluate the impact of obesity and TRF on p-AKT expression. $N = 3$ independent experiments from dissected IFMs from ten female flies per genotype. **i** Nile Red stained lipid bodies of 3-week-old female ALF flies exhibited larger and more irregular shapes compared with TRF. Scale bar is 20 µm. Nile red quantification of lipids in the adipose tissues (**j**) in 3-week-old female fly IFMs showed upregulation of fat contents in obese models and suppression under TRF. $N = 9$ from three specimens each for panels **f** and **j**. Bar graphs (along with dot plots for panels **b**, **c**, **d**, **f**, **h**, and **j**) and error bars presented as mean ± SD and statistical analysis was performed using one-way ANOVA with post hoc Tukey test, where * $p < 0.05$; ** $p < 0.01$; *** $p < 0.001$; ns   nonsignificant

TRF reduced expression of *Nlaz* RNA in skeletal muscle in both obesity models, with a reduction of ~40% for both WT-HFD and *Sk2*-RD (Fig. 6d).

To investigate the impact of TRF on IMCL, we examined the IFM (Fig. 6e, f). The degree of lipid infiltration in the IFM was exacerbated in both DIO and GIO and was reduced under TRF (Fig. 6e, red puncta). Furthermore, TRF improved myofibril organization, and reduced the number of gaps and tears compared with ALF (Fig. 6e). The total lipid area in the IFM decreased under TRF in wild-type, HFD and *Sk2* flies, compared with age-matched ALF flies (Fig. 6f).

Elevated p-AKT has been associated with insulin resistance in mouse skeletal muscle and liver tissues under obesogenic challenges[41]. Compared with age-matched WT, both DIO and GIO resulted in significant upregulation of the p-AKT level detected by phospho-Drosophila AKT (Ser505) antibody (Fig. 6g, h). Interestingly, the p-AKT level was significantly reduced under TRF condition in both DIO and GIO, compared to age-matched samples under ALF (Fig. 6g, h).

*Drosophila* fat bodies are considered equivalent to vertebrate adipose tissue, both metabolically and in their endocrine role[42]. Lower abdomen sections of fly fat bodies showed that both DIO and GIO resulted in a greater proportion of lipid droplets with aberrant shapes and increased size (Fig. 6i, white arrows). TRF decreased the number of abnormally shaped droplets, as well as their average size, by nearly half ($\geq 49\%$) in both WT-HFD and *SK2*-RD flies (Fig. 6i, j). TRF also mitigated lipid deregulation associated with age in wild-type flies (Fig. 6i, j; reduction of 32%).

Resistance to abiotic stressors, such as exposure to harsh temperatures and prolonged starvation, is associated with longevity and general resilience[43]. We tested if adherence to TRF affects survival under 24 and 48 h of starvation in 3-week-old flies. TRF increased survival in multiple comparisons, particularly at 48 h (Supplementary Fig. 1c, d). We also tested whether TRF lengthens lifespan (Supplementary Fig. 2). Surprisingly, TRF did not result in the expansion of median lifespan in wild-type flies fed RD, DIO and GIO.

**TRF suppresses obesity-induced myofibril and mitochondrial defects**. We also investigated the impact of TRF on the IFM at the ultrastructural level. Both DIO and GIO models exhibited myofibrillar disorganization and mitochondrial abnormalities (Fig. 2, Fig. 7a; Supplementary Fig. 4). Interestingly, these phenotypes were attenuated under TRF compared with age-matched ALF 3-week-old female flies (Fig. 7b). TRF suppressed the disruption of the M-lines and Z-discs (along with other myofibril disorganizations) and improved mitochondrial abnormalities

associated with obesity (Fig. 7; Supplementary Fig. 1f). Most EM images of both DIO and GIO under TRF showed strikingly similar profiles to age-matched wild-type control under ALF or TRF (Fig. 7a, b; Supplementary Fig. 1f). Together, our findings reveal obesity-induced myofibrillar and mitochondrial abnormalities, and the potential for the TRF paradigm to prevent these abnormalities at the ultrastructural level.

**TRF restores muscle dysfunction under circadian rhythms disruption**. It is estimated that ~20% of the US workforce is exposed to abnormal light conditions due to shiftwork[44,45], which is known to compromise cardiometabolic fitness. We have shown that light-induced circadian disruption compromises cardiac performance[46]. To evaluate the impact of TRF on skeletal muscle function when circadian rhythms were disrupted by light, we placed flies (*Canton-S*, *w1118*, or *SK2* mutants) in constant light (LL) and fed them RD under ALF or TRF conditions (Fig. 8a). These strains exhibited reduced flight performance, even within the first week, compared with flies reared in light/dark (LD) conditions (i.e., under normal circadian rhythm conditions) (Fig. 8b–d). Interestingly, for most strains (particularly older flies), TRF increased flight performance for one or both sexes (Fig. 8b–d). Exposure to constant light (LL) reduced geotaxis performance throughout the length of the experiment (Fig. 8e–g) compared with age-matched LD, yet TRF prevented declines in locomotion, particularly in older animals (Fig. 8e–g).

We stained cryosections of IFMs from middle-aged flies to evaluate myofibril integrity and ectopic fat deposition. Like diet and genetic models of obesity, light disruption led to the formation of intramuscular lipid deposits and myofibrillar disorganization (Fig. 8h). TRF ameliorated these conditions, even in obesity and light-challenged *Sk2*-LL flies. Adherence to TRF reduced ectopic fat deposition by nearly half ($\geq 41\%$) in both WT and *Sk2* mutants under LL (Fig. 8i). Interestingly, TRF reduced the expression of *NLaz*, suggesting that TRF remains effective in ameliorating metabolic dysfunction even in LL (Fig. 8j).

**Discussion**

There is an urgent need to model obesity and metabolic diseases in laboratory animals to test the efficacy of novel lifestyle interventions and alleviate these disorders. This study achieves these goals using a *Drosophila* model of metabolic disease. First, we modeled three principal causes of obesity and metabolic diseases; GIO, DIO, and CRD. Second, we identified these three to be causes of metabolic diseases that compromise physiology of the skeletal muscle, which is the largest metabolic organ in

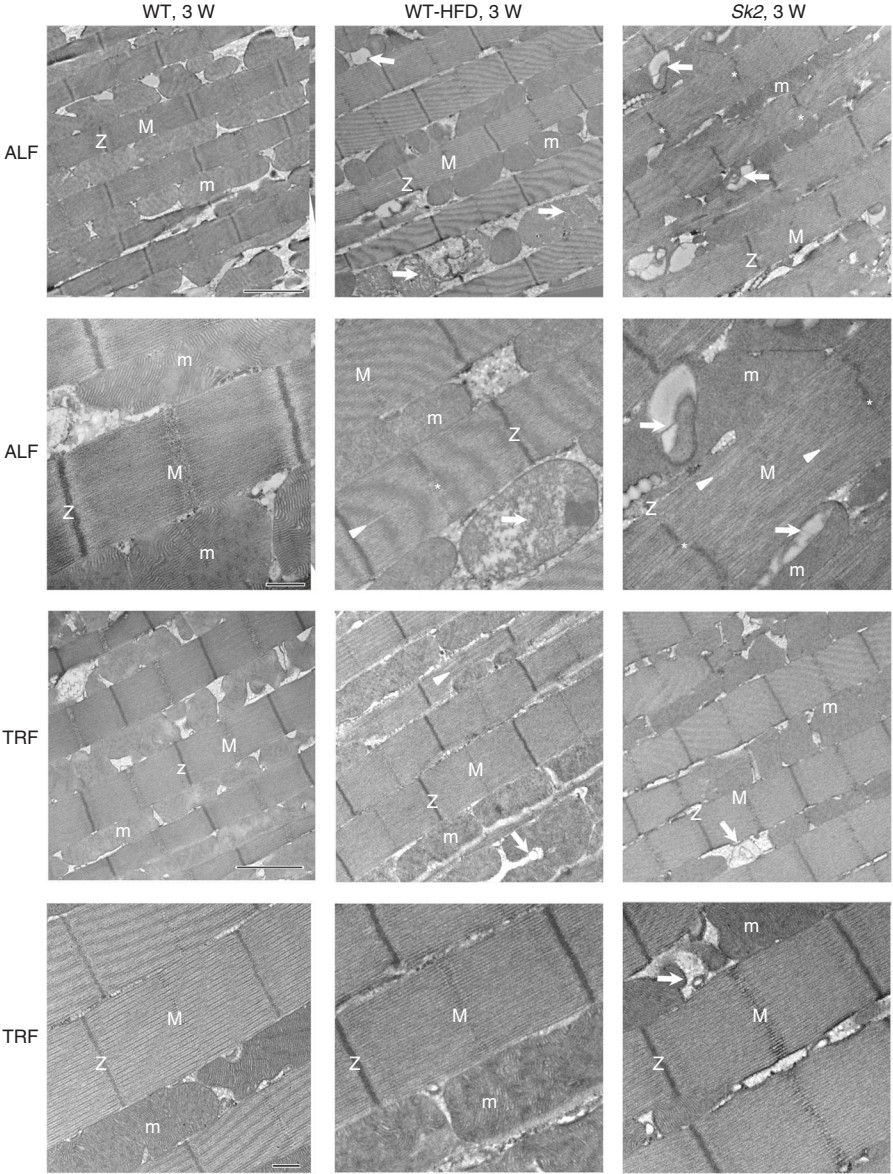

**Fig. 7** TRF suppressed obesity-induced myofibrillar and mitochondrial defects at the ultrastructural level. Ultrastructural analysis using TEM revealed that obesity caused myofibril disorganization and mitochondrial abnormalities compared with wild-type CS under ALF in 3-week-old female flies IFM. Compared with control (left top two panels), occasional myofibrils with partially intact Z-disks (Z) and M-lines (M) were observed (middle top two panels, asterisks and arrowheads) with the diet-induced obesity which were more prominent in the genetic obesity mutant (right top two panels, asterisks and arrowheads). The abnormal mitochondria (m) were evident with diet and genetic obesity partially or completely degenerated cristae (arrows). Scale bars for the top and bottom panels under ALF are 2 μm and 0.5 μm, respectively. Myofibrillar and mitochondrial abnormalities were significantly ameliorated under TRF in both obesity models. Scale bars for the top and bottom panels under TRF are 2 μm and 0.5 μm, respectively. Most of the images of the diet and genetic obesity model look like wild-type under TRF. The quantification of myofibrillar and mitochondrial associated with obesity are shown in supplementary Fig. 1f

humans and animals. Finally, we found that TRF can improve muscle function and lessen the severity of metabolic diseases arising from three principal drivers.

Impaired muscle function and muscle insulin resistance are often associated with human pathological obesity. In this study, we found that *Drosophila* subjected to HFD and HSD progressively impaired flight index and geotaxis activities which offer a physiological readout of muscle function. This impairment was associated with excessive IMCL deposition in muscle. IMCL accumulation alone may not impair muscle function, as regular IMCL accumulation and regular depletion during intense

physical activity is also observed among endurance trained athletes. However, gradual and tonic accumulation of IMCL may ultimately impair muscle ultrastructure that paralleled impaired flight index and geotaxis movement. In addition, ectopic fat is a hallmark of insulin resistance in diabetes, and its role in obesity comorbidities is thought to be mediated by the cytotoxic effects of lipid deposits and resulting metabolic changes in those tissues[47,48].

Investigation into the mechanism of DIO has led to the identification of key metabolic regulators. Animals carrying hypomorphic or loss-of-function alleles of these genes also replicate

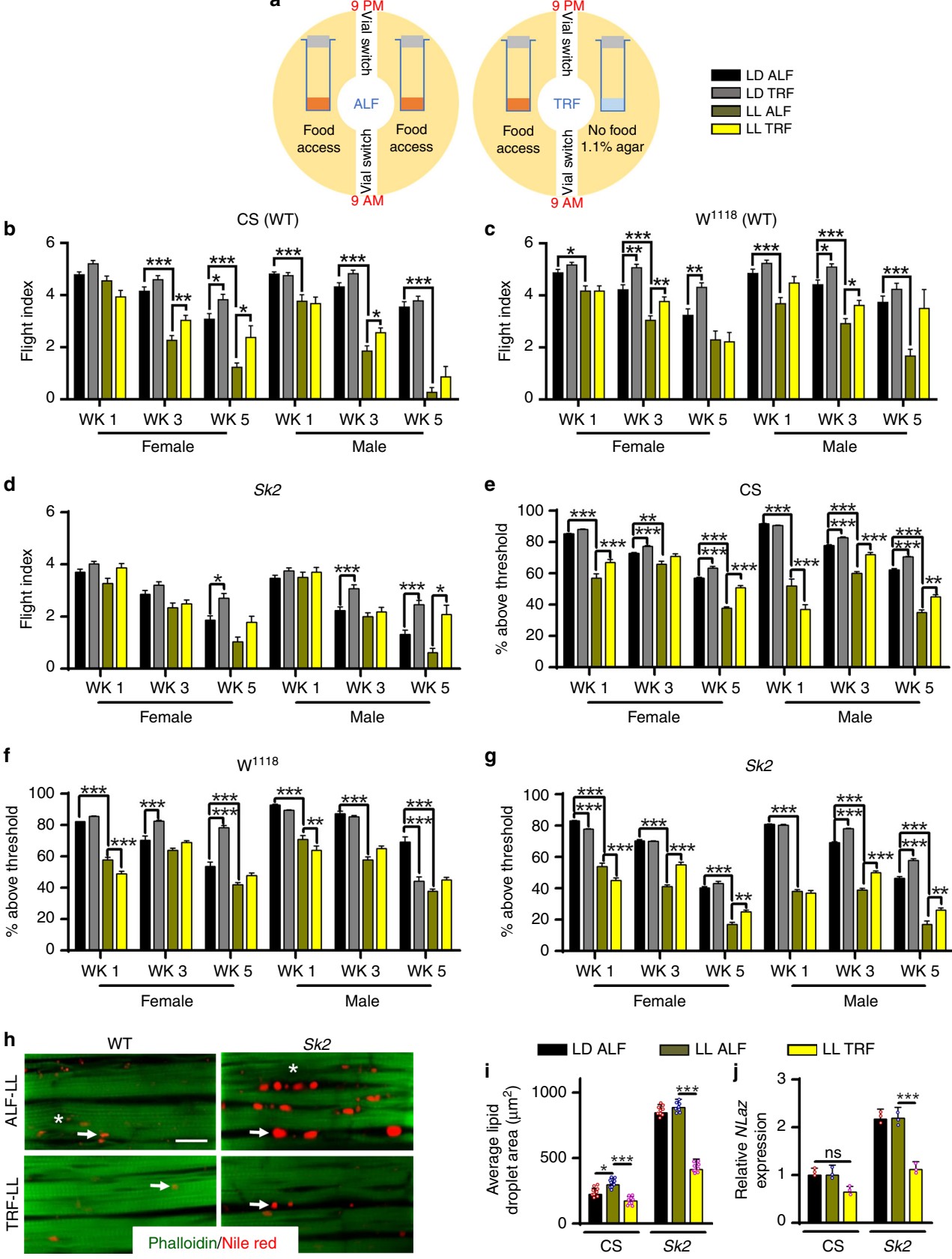

**Fig. 8** TRF suppresses skeletal muscle dysfunction and intramuscular lipid infiltration produced by circadian light disruptions. **a** Displaying light-induced circadian disruption. Flies were placed in either LD (light/dark, 12 h cycles; see Fig. 3a) or LL (constant light conditions) and either given unrestricted food access, *ad libitum* (ALF), or allowed to eat only during the daytime (TRF, see the Methods section). **b–d** LL flight performance. Both male and female flies were tested at 3-time points to characterize early, mid, and late stages of skeletal muscle performance. LD data relabeled from Fig. 2d–f. Statistical comparisons: ALF vs TRF and ALF LD vs ALF LL. Bars represent 10–12 separate replicates tested per cohort. $N = 90$ per condition. **e–g** Climbing performance was measured using the geotaxic assay (Fig. 2c) at the aforementioned time points using the same statistical comparisons as g, h. $N = 90$ per condition from 10–12 separate replicates tested per cohort. **h** Confocal images of the IFM revealed that TRF remains effective in reducing ectopic lipid deposition in wild-type and obese flies even in light-induced circadian disruption. Scale bar is 20 μm. **i** Nile red intramuscular lipid area quantification. Fat infiltration was reduced with adherence to TRF in 3-week-old female LL flies. $N = 9$ from three specimens each. **j** TRF reduced the expression of *NLaz* in the flight muscles of 3-week-old flies under LL. Expression normalized to RPL11. CS set to 1. $N = 3$. Bar graphs (along with dot plots for panels **i** and **j**) and error bars data presented as mean ± SD and statistical analysis was performed using ANOVA with post hoc Tukey test for panels (**b–d**, **i–j**) and post ad hoc Sidak's method with multiple comparisons for panels **e–g**, where * $p < 0.05$; ** $p < 0.01$; *** $p < 0.001$; ns nonsignificant

various features of DIO even when they are fed a standard diet. In this study, we found flies carrying hypomorphic or loss-of-function alleles of three such genes *Bmm*, *Sk2*, and *Ifc* replicate various features of pathological obesity. Specifically, they exhibit physiological impairment of muscle function which exacerbates upon ad lib feeding of obesogenic diet (Fig. 1). Although genetic basis of obesity is not as prevalent in humans as DIO, it is likely that weaker alleles of obesogenic genes interact with an energy dense diet to exacerbate pathological obesity and metabolic diseases (Figs 5–7).

An emerging cause of metabolic disease appears to be circadian rhythm disruption, which can arise from prolonged exposure to light, shiftwork, or shiftwork-like lifestyle[44,45]. A recent estimate suggests as much as 70% of adult population in the western world experience at least few years of shiftwork-like style by the age of 40 years[49]. Previous studies have shown CRD in flies could impair cardiac physiology, and reproductive function by adversely impacting nutrient metabolism[9,10,50]. We found rearing flies under constant light (LL), a known disruptor of circadian rhythm also impairs muscle function as revealed by reduced flight index and geotaxis (Fig. 8).

Importantly, epistatic interaction among DIO, GIO, and CRD also revealed at least additive effect between any two factors. For example, fly models of GIO when fed an obesogenic diet showed more severe metabolic defects than DIO or GIO alone (Fig. 5). Similar interactions were also seen between DIO and CRD or GIO and CRD (Figs 1, 5, 6, 8). In other word, making better diet choices can reduce the severity of metabolic diseases arising from genetic risk. But this benefit of a standard diet (relative to that with an obesogenic diet) on reducing adverse impact of GIO can be blunted by CRD (Fig. 8). The interaction between GIO and CRD found in this study complement some of the rodent studies. In rodents, genetic defects increasing the risk for liver disease in combination with CRD further exacerbates liver disease and increase the risk for liver cancer[51]. In this study, we found CRD exacerbates genetic risk for metabolic dysfunction and compromise muscle function (Fig. 8).

For the first time, we systematically evaluated the effect of TRF on reducing the adverse effects of DIO, GIO, and CRD on muscle function. Several studies have shown the efficacy of TRF in both preventing and reversing metabolic diseases in rodent models of DIO[26,27,52,53]. The underlying complex mechanisms involve improved liver function, reduced hepatic triglyceride deposit, increased brown fat activity, and reduced adipose tissue inflammation[26,27,52,53]. In this study, we found TRF reduces the adverse impact of DIO by reducing lipid deposit in muscle, reducing insulin resistance, sustaining muscle function, and sustaining ultrastructural integrity of muscles (Figs 5–7). Such overall positive impact of TRF on muscle appears to be conserved in vertebrates as rodents under TRF also exhibit improved endurance and improved motor coordination[26,27,52].

The impact of TRF on genetic obesity is not well studied. Mice lacking normal function of circadian clock are known to be prone to obesity or metabolic diseases[54]. However, TRF prevents obesity and metabolic diseases in circadian clock mutant mice[52,55,56]. This observation raised the hypothesis that the function of the circadian clock is to support a daily cycle of feeding and fasting. This cycle supports the cyclic activation and repression of nutrient- and fasting-sensing pathways, which in turn regulate the temporal activation of anabolic and catabolic pathways for nutrient utilization. Fatty acid and TG metabolism in wild-type mice are strongly circadian. Therefore, hypomorphic mutant of these pathways may compromise normal flux through these pathways during fed or fasted state. ALF of a diet, and specifically an energy dense diet, disrupts the normal function of nutrient sensing mechanisms by unknown mechanism. This results in animals exhibiting molecular markers of fasting and feeding being constitutively activated throughout 24 h day[26,27,52]. Consequently, enzymes of anabolic or catabolic pathways that are normally required only during fed or fasted state are tonically activated throughout 24 h. Such temporal disruption of nutrient sensing pathways in combination with hypomorphic mutants of nutrient metabolism further exacerbates dysmetabolism and leads to pathological obesity. TRF substitutes the function of circadian clock by imposing a consistent cycle of feeding and fasting and restores normal cyclic activation and repression of nutrient sensing pathways. Accordingly, when mutant animals are subject to TRF, the function of hypomorphic mutants in anabolic or catabolic pathway is not required for 24 h, but only for the duration of feeding or fasting. This may lessen the genetic burden and lessen the genetic risk for metabolic disease.

TRF also reduces the risk for compromised muscle function in flies held under continuous light and adds to a growing interest in adjusting mealtime as an effective approach to cope with shiftwork. In a rat model of jetlag or shiftwork, TRF aligns with the nocturnal active phase of the animal that rapidly re-entrains the peripheral circadian clocks[57]. This can reduce desynchronization between clocks in the neural and the peripheral tissues and may improve metabolic health. However, the LL paradigm used in our study likely disrupts both neural and peripheral clocks. While TRF is known to sustain the peripheral clocks, it is not clear if TRF restored some rhythms in the CNS clock. Therefore, the physiological benefits under TRF in flies held under LL may arise from sustained rhythms even in the presence of a disrupted CNS clock. Such partial restoration of rhythms only in peripheral organs may explain the reduced benefits of TRF in LL flies relative to that observed in flies held under LD (Fig. 8).

This study also highlights important differences between *Drosophila* and rodent models of TRF with respect to daily duration of TRF, the role of circadian clock and effect on lifespan. TRF of 8–12 h without reducing caloric intake are feasible in rodents and rodents show increasing benefits of TRF for body

weight, adiposity, and endurance with shorter duration of TRF[58]. However, eating duration of < 12 h is not well tolerated in Drosophila, making it impossible to test the impact of a shorter duration TRF in insects. A recent study revealed that the most widely used method to do caloric restriction (CR) in rodents also involve time-restricted feeding of < 6 h and pair-feeding or meal-feeding protocol involves TRF of 12–13 h[59,60]. While it is well known that CR extends lifespan in rodents, meal-feeding protocol that mimics 12–13 h TRF also moderately extends rodent lifespan[60,61]. However, in our study, 12 h of TRF failed to extend lifespan, when the flies modeled DIO or GIO. CR protocols in flies that increase lifespan involve calorie dilution and TRF imparts a gene expression signature that is distinct from CR in flies[14]. Given that flies cannot tolerate a shorter duration of feeding that can potentially reduce caloric intake, it remains inconclusive to directly compare the lifespan extension effect of CR and TRF in flies.

Some of the metabolic benefits of TRF in rodents are independent of a functional circadian clock[52], while a previous study in flies had shown some of the cardiac benefits of TRF requires a functional circadian clock[9,10]. Flies lacking some of the circadian clock components also exhibit cardiac defects which are like cardiac defects found in heart-specific circadian mutant mice[62]. Altogether, these results imply further studies are necessary to examine the optimal combination of quality, quantity, and timing of nutrition in different genetic backgrounds that can ultimately contribute to personalized nutrient optimization to extend both healthy lifespan and total lifespan.

## Methods

**Drosophila models, diets, and feeding fasting regimines**. Fly strains used Canton-S (CS), w[1118], Sphingosine kinase 2 (Sk2, Bl #14133), Infertile Crescent (Ifc, BL#1549), and Brummer (Bmm, Bl #15828) were obtained from Bloomington (BL) Drosophila Stock Center[17,19,20]. Homozygous Sk2 mutants were used; however, homozygous lethal Ifc and Bmm mutants crossed with w[1118] were retained as viable heterozygotes with balancer chromosomes. Standard regular diet (RD): agar 20 g/L, sucrose 80 g/L, active dry yeast 5 g/L, calcium chloride dihydrate 1.6 g/L, ferrous sulfate heptahydrate 1.6 g/L, sodium potassium tartrate tetrahydrate 8 g/L, sodium chloride 0.5 g/L, maganese chloride tetrahydrate 0.5 g/L, nipagen 5.3 mL/L. High-fat-diet (HFD): standard diet supplemented + 5 % coconut oil[10]. High-sugar diet (HSD): standard diet supplemented + 300 mM Sugar[63]. Mixed high-fat and sugar diet (HSD + HFD): standard diet supplemented + 5% coconut oil and 300 mM sugar. One-day-old wild-type and obese flies were collected and kept at 22 °C, 50% humidity in a 12 h light/12 h dark (LD) cycle[10]. Alternatively, flies were maintained in constant light conditions (LL)[46]. Adult flies were collected upon eclosion, separated by sex, and maintained on a standard diet in groups of 20–25 for 3 days. Vials were assigned a diet, feeding regimen and light treatment on the fourth day. Flies were transferred onto fresh media every day. ALF and TRF flies were switched to either a new food media vial or a 1.1% agar, respectively, at zeitgeber time zero (ZT 0) 9 PM (lights off), and were returned to their original vials the day after at 9 AM (lights on) (Fig. 2a)[10]. To evaluate muscle-specific function of the Sk2, RNAi stocks were obtained with VDRC. We also generated a recombinant stock using multiple genetic crosses[14] possessing Fln-Gal4 (flightin gene, an IFM-specific driver) and UAS-mito-GFP, which labels the mitochondria with GFP[16]. Progeny of UAS-Sk2 RNAi and recombinant Fln-Gal4:UAS-mitoGFP were collected for the physiological and cytological studies. To generate heterozygous Sk2, Bmm, and double-mutant standard, genetic technique was used, and progeny without any markers/balancer were used for the physiological and cytological studies[14]. Physiological parameters were measured at 1, 3, and 5 weeks of age from eclosion. Both virgin male and females were collected but kept separate throughout their lifespan.

**Feeding, body weight, triglyceride, and glucose**. Capillary feeding assays (CAFE) were carried out as illustrated in Supplementary Fig. 1b[10]. Briefly, flies were setup in empty 25- mm vials (1 fly per vial) with a cotton piece soaked with 1 ml of water placed at the bottom of the vial to maintain humidity. Multiple chambers were setup with the calibrated glass micropipette (5 µl, catalog no. 53432-706; VWR, West Chester, PA), with or without flies. Five micropipettes per chamber were setup by inserting through top cap, and filled by capillary action with liquid medium containing 5% (wt/vol), sucrose 5% (wt/vol) autolyzed yeast extract (Bacto yeast extract; BD Diagnostic Systems), and 0.05% (wt/vol) Blue dye #1 (ThermoFisher Scientific). Five male or female flies (3-week-old wild-type, DIO and GIO) were placed per food chamber for 24 h before collecting data for 3 days. One chamber containing five micropipettes was setup without flies to test daily

evaporation. Daily food, day and nighttime food consumptions from each chamber, and micropipette containing flies and those without flies were calculated. Average food consumption was calculated by subtracting average evaporated food from the pipettes that do not have flies from the micropipettes containing flies. Different in daily, day and night food consumptions between control and obesity groups was calculated[10]. Both ALF and TRF flies had access to capillaries with food solution during the daytime, while at night, only ALF had capillaries with food. At night, TRF flies had access to moisture from the damp cotton piece. For body weight measurements, both ALF and TRF flies were transferred to 1.1% agar vials overnight. 30–50 flies were anesthetized and weighted in tared tubes from minimum triplicate experiments[10]. The total triglycerides in the IFM levels were determined using Infinity TG Reagent kits (Sigma) as per kit instruction[17]. For the hemolymph glucose and trehalose, hemolymph was extracted from minimum 50 flies per genotype[7,33,64]. Briefly, hemolymph was pooled from minimum 50 flies per genotype from the thorax by capillary action. Hemolymph was diluted 1:10 before assay. Glucose was measured by adding to 99 µl of Thermo Infinity Glucose Reagent (TR15321) in a 96-well plate, then incubated for 3 min at 37 °C and read immediately at 340 nm. Trehalose content was also determined using per kit instruction[40]. Sampling for the all the hemolymph glucose quantification was carried out at zeitgeber time 12 (ZT 12), during daytime (9 AM) when light was on in both ALF and TRF flies. Both sets of flies ALF, and TRF, which were transferred to agar vials for 12 h before sampling[10].

**Real-time quantitative PCR**. Neural Lazarillo (NLaz) expression was carried out using RT-qPCR from micro-dissected indirect flight muscle (IFM)[14] and as indicated for hemolymph glucose assay, all the sampling for the NLaz quantification was carried out at zeitgeber time 12 (ZT 12), during daytime (9 AM) when light was on in both ALF and TRF flies[10]. A minimum of three independent biological samples for each condition was used from 3-week-old flies for one of the diet-induced obesity conditions (CS + 5% fat) and one of the genetic obese mutants (Sk2). Briefly, dissected IFM was placed in RNA lysis buffer, and flash frozen. RNA was extracted using RNeasy mini kit (QIAGEN) with on column DNase digestion. Quantitative RT-PCR was done following standard protocols. Expression was normalized with 60S ribosomal protein L11 (RPL11)[10]. Analysis was done using the absolute quantification method Relative mRNA levels of non-injured flies were set at 1.0, and subsequent expression levels from different time points were expressed as normalized values[10]. Primers for qPCR are listed in Supplementary Table 3.

**Western blot analysis**. AKT and Phos-AKT protein levels were determined by performing western analysis on protein extracts from dissected IFM from 3-week-old adults[65,66]. Briefly, ten thoraxes were added to 100 µl of lysis buffer (62 mM Tris pH 7.5, 2% SDS with protease and phosphatase inhibitors)[14]. The samples were boiled for 5 min at 95 °C and centrifuged to collect the supernatant. Sample loading buffer was added and then the mixture was boiled for 1 min at 95 °C. The samples were then electrophoresed on 10% AnykD Mini-PROTEAN TGX Precast Gels (Bio-Rad). The protein bands were transferred to the PVDF membrane (Bio-Rad) and incubated separately with Phospho-Drosophila Akt (Ser505) antibody, catalog # 4054 (1:1000, Cell Signaling), Akt antibody, catalog # 9272 (1:1000 dilutions, Cell Signaling) and beta-actin antibody, catalog # 8457 (1:1000 dilutions, Cell Signaling). For the secondary antibody, horseradish peroxidase-conjugated anti-rabbit IgG was used at 1:1000 dilutions. SuperSignal West Pico chemiluminescent substrate was used to detect the signal, and the corresponding band was quantified using ImageJ software[14]. Western analysis was performed on three independent biological replicates for each genotype under ALF and TRF. An uncropped and unprocessed scans probed for the p-AKT is shown in the Supplementary Fig. 5 for the wild-type and both obesity conditions under ALF and TRF.

**Bromoenol lactone (BEL) treatments**. For the pharmacological blocking of the ATGL, BEL (Cayman Chemicals) was carried out. Briefly BEL was dissolved in ethanol and added in the food (50 µM final concentrations). For the control experiment, same amount of ethanol without BEL was added in the food. Adult flies (control, HFD, and Sk2 mutant) were kept in the food with BEL or without BEL for 1 week upon eclosion and used these flies for functional and cytological studies as described below.

**Flight muscle performance**. For skeletal muscle functional defects linked with obesity, we have evaluated muscle performance via flight test (Fig. 5b). This methodology[13,14,67] involves release of control and experimental adult flies into the center of a Plexiglas box with a light source positioned at the top. Based upon each animal's ability to fly up [6.0], horizontally [4.0], down [2.0], or not at all [0.0], we assign flight indexes (FI) for different ages and genotypes of flies. The average FI was calculated by dividing the sum of the individual FI values by the number of individuals for each group. Obesity-induced and age-dependent FI (both male and female, 100–300 flies each) were evaluated in 1-, 3-, and 5-week-old flies along with the respective control. The impact of TRF in ameliorating obesity-induced skeletal muscle dysfunction was also evaluated.

**Negative geotaxis**. The geotaxis assay was carried out to revel locomotion defect linked with obesity and impact of TRF on locomotion (Fig. 5c)[68]. Briefly, 100–150 flies at 1-, 3-, and 5-week-old ages were transferred to a new vial (10 flies per experiment) and allowed to rest for 2 min to acclimatize. The vial was then sharply tapped three times to stimulate a negative geotaxis response. Climbing ability of the flies was video-recorded and saved for data analysis. At 10-s intervals, the fraction of flies that climbed to a 7-cm mark was calculated.

**Cytological analysis of muscle and adipose tissues**. For the cytological assay, the fly head, legs, and wings were removed and fixed in 4% PFA solution for 20 min[14]. The fixed samples were washed three times in PBS with a 15-min incubation time between each wash. The samples were then aligned longitudinally (thoraces) in a cryomold filled with Tissue-Tek OCT (Sakura), and then flash frozen in dry ice. The samples were cryosectioned at 30 -μm thickness, and the samples were washed in 1× PBS thrice with 15-min incubations between each wash. For structural aberrations, we used 0.1 -μm Alexa594-Phalloidin for 30 min before washing three times in PBS to detect actin-containing myofibrils. To detect obesity-induced lipid misregulation, cyrosectioned samples were stained with Nile red[69]. Briefly, Nile red (ThermoFisher scientific) at 4 μg/mL in 75% glycerol was diluted to 1:4 in 25% glycerol in water then applied to the sample for 30 min and washed with PBS three times[69]. Confocal images were taken using a Zeiss 710, and the quantification of lipid droplets was performed using ImageJ as previously described[14]. For the MitoTimer staining, we generated a new line using genetic recombination *UAS-Mito-Timer* (Bl # 57323) with *Mef2-Gal4* driver. We crossed that line with *Sk2* or CS, and the progeny were collected and aged for 3 weeks. Cyosections of IFM were produced as described above, and confocal images taken with a Zeiss 710[14]. For analyses using *mito-GFP*, a recombinant stock possessing *Fln-Gal4* (*flightin* gene, an IFM-specific driver) and *UAS-mito*-GFP, which labels the mitochondria with GFP[16] was produced. This stock was crossed with *UAS-Sk2 RNAi*, and progeny were collected and aged. Muscle samples of 3-week-old flies were fixed as described above, stained with 0.1 μM phalloidin (TRITC), and mounted with the vectashield mounting medium containing DAPI. Imaging was performed using a Zeiss 710 Confocal Microscope. Quantification of mitochondrial dysmorphology/fragmentation was performed using ImageJ[14].

**Ultrastructure analysis of muscle**. Transmission electron microscopy (TEM) was performed on IFMs of 3-week-old females using at least three different samples each for the control and obese models. Sample preparation was carried out using primary fixative (3% glutaraldehyde, 3% paraformaldehyde, 0.1 M sodium cacodylate buffer pH 7.2, in 2 mM EGTA relaxing buffer), overnight on ice[13,14,70]. Samples were washed 6 ×, 30 min each in 0.1 M sodium cacodylate buffer, pH 7.2 on ice, then treated with secondary fixative (1% OsO$_4$, 0.1 M sodium cacodylate buffer pH 7.2, and 10 mM MgCl$_2$) for 2 h on ice. All subsequent steps were performed at room temperature. Samples were washed 3× for 10 min each in 2 -mL HPLC grade H$_2$O, then dehydrated in an acetone series (25%, 50%, 75%, 95%, and 3 × 100% anhydrous acetone) for 30 min each. Samples were infiltrated in Epon mix [16.2 mL of EM bed-812, 10.0 mL of dodecenyl succinic anhydride, 8.9 mL of nadic methyl anhydride, 0.6 mL of *2,4,6-Tris(dimethylaminomethyl)phenol*] for 2 h each using increasing ratios of Epon mix:dry acetone (1:3, 1:1, 3:1). Following infiltration in 100% Epon mix for 16 h, samples were polymerized in Epon-filled BEEM capsules at 60 °C for 1 day under vacuum. Longitudinal sections of IFMs were obtained via ultramicrotome and stained with 2% uranyl acetate. TEM analysis was used to determine the impact of obesity on myofibrillar organization, cytoskeletal integrity, and architecture of the mitochondria[13,14,70]. All the TEM images under ALF were obtained using a FEI Tecnai 12 transmission electron microscope with a TVIPS 214 high-resolution camera at the SDSU. In addition, TEM images under TRF and ALF regimen for the control and obesity condition were obtained at the Salk using a Zeiss Libra 120 PLUS EF-TEM, operated at 120 kV with a YAG CCD 2k camera at full resolution with Zemas Data Collection (Appfive, LLC) imaging software. Quantification of myofibrillar and mitochondrial abnormalities (a minimum of 150 sarcomere and mitochondria per condition from three flies were used for quantification) was carried out using ImageJ[14]. Briefly, for the evaluation of intact sarcomere, M-lines and Z-discs, two criteria were evaluated. (1) The M-line and Z-disc were scored as intact if they were continuous along the sarcomere width. (2) The alignment of the M-lines and Z-discs within the sarcomeres was determined by drawing a straight line on each using ImageJ. Similarly, percentage of the intact mitochondria were evaluated using ImageJ[14].

**Lifespan, activity, and starvation survival**. Flies were collected and maintained as described. Approximately 30 flies were placed in each vial and transferred to a new vial every day. The numbers of surviving adults were counted every day. This was compared with the original number of adults collected on day zero, and the percentage for each day was graphed. Survival was determined and the median lifespan was defined as 50% survival[69]. Log-rank test was used to determine statistical significance. Activity of 3-week-old wild-type and obesity flies was assessed in a TriKinetics DPM (*Drosophila* population monitoring) system[10]. Daily food changes under ALF and were performed as described above. Reported data are daily averages of at least 7 days of monitoring. For the starvation assay, flies under ALF

and TRF were starved for 24 or 48 h at 22 °C in 1.1% agar medium and survivors were counted at the end of the assay.

**Statistical analysis**. For most quantitation except lifespan analysis, one-way ANOVA with post hoc Tukey test with multiple comparisons was used to determine significance. The remaining statistical test (5d, f, g, and i red asterisks only; 8i and 8j) used ANOVA with post ad hoc Sidak's method with multiple comparisons. All tests performed were done using GraphPad Prism 6[10,14,46]. Bar graphs show mean and standard deviation (SD), written as mean ± SD, with significance presented as $*p < 0.05$, $**p < 0.01$, $***p < 0.001$. In addition, Kaplan–Meier used $****p < 0.0001$ with statistics calculated using Log-rank (Mantel–Cox) test to determine significance between survival curves. Significant differences were assumed at $p < 0.05$.

**Reporting summary**. Further information on research design is available in the Nature Research Reporting Summary linked to this article.

## Data availability
Additional supporting data of this study include added metabolic, cytological, ultrastructure, and lifespan data. Additional supporting data also include extended methodology examples and statistical analyses. The data that support the findings of this study are available from the corresponding author upon reasonable request.

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

## Acknowledgements

We would like to thank Dr. Sanford Bernstein, San Diego State University, for allowing us to use his lab facility, and Jennifer Suggs for help with the EM preparation. We also like to thank Andrew Han and Jay Vyas, San Diego State University for their help with genetic crosses and maintaining stocks. We would also like to thank Casey Peto, Salk Institute, for help with EM preparation and imaging, and Terry Lin for the help with immunoblotting. We also would like to thank Dr. David O'Keefe Salk Institute, for his editorial comments on the manuscript. This work was supported by National Institutes of Health (NIH) grants AG049494 to GCM and DK115214 to SP. This work was also supported by the Waitt Foundation and Core Grant applications NCI CCSG (CA014195) and NINDS Neuroscience Center (NS072031) to LA and UM. AST is a Fellow of the Rees-Stealy Research Foundation and the San Diego State University Heart Institute. The content is solely the responsibility of the authors and does not necessarily represent the official views of the NIH.

## Author contributions

G.C.M. designed the experiments, analyzed the data, and prepared the paper with the help of J.E.V. J.E.V. and C.L. performed most of the experiments with assistance from S.C., B.W. and H.D.L. A.S.T., L.A., and U.M. performed ultrastructural studies and prepared EM figures, and edited the paper. S.P. provided advice on experiments and assisted with the paper preparation and edited the paper.

## Additional information

**Competing interests:** The authors declare no competing interests.

