## [Peer Review File · Nature Communications]

Reviewers' comments:

Reviewer #1 (Remarks to the Author):

The overall theme of this manuscript is that restricting time of feeding (TRF) as an intervention to improve markers of metabolic disease in both diet and genetic obesity models in the fly. Additionally the authors argue that striated muscle is a major pathological target in obese flies and that improving muscle health with TRF is a significant contributor to systemic health. The topic is an exciting area for research related to metabolic diseases.

The major observations reported by the authors of this paper include:

Figure 1: Establishment of the diet induced and genetic induced models of drosophila obesity with concomitant changes weight and indices of flight/climbing.

Figure 2: Time restricted feeding attenuates the decline in motor behavior of the Sk2 mutant flies in response to high fat feeding or aging.

Figure 3: Time restricted feeding rescues increased lipid, a read out of insulin resistance in models of obesity

Figure 4: Time restricted feeding rescues metabolic profile in muscle when flies are housed in constant light (model of circadian disruption).

The primary concerns are the following:

- 1) In general the outcomes assays are very descriptive and do not directly test either muscle function or metabolic sensitivity. This limits the interpretations.
- 2) The authors can conclude that time of feeding improves indices of flight and climbing and some histological aspects of muscle but there are no true muscle function assays in this study. Any reference to improved muscle function must be softened and acknowledged that the read out is downstream and there could be other factors (i.e. innervation) that might be altered with TRF
- 2) It is unclear why there are some experiments with n=3 while other experiments have n=30-100 (see the legend for Figure 1)? There are large discrepancies in sample number across experiments and this needs to be cleared up.
- 3) Expression of NLaz mRNA is used as a marker of insulin resistance. It is not clear how sensitive this marker is? It is not clear why the authors did not measure glucose or more direct measures of changes in glucose transport/metabolism to more clearly define the metabolic disease aspect of these models.
- 4) A minor point is that the use of the Sk2 fly strain for obesity should be referenced in the Introduction. References should be checked through out the manuscript.

Reviewer #2 (Remarks to the Author):

In this manuscript, the authors set out to test the effectiveness of time-restricted feeding as a therapy for improving skeletal muscle physiology and function in the context of genetic (GIO) or diet (DIO) induced obesity models in *Drosophila*. Previous studies in both mammals and flies have established that the negative health outcomes of high-fat diet and obesity can be rescued if organisms are restricted access to food during their respective active phases. In the fly, TRF has been demonstrated to ameliorate age-dependent declines in cardiac muscle, but to date, there does not appear to be a functional characterization of the effects of TRF of obesity models on skeletal muscle physiology and function with age. The main findings of this paper are that GIO and DIO lead to increases in lipid droplet formation within skeletal muscle (IFC), decreases in flight index, and declines in climbing activity which are protected with the TRF paradigm.

From these findings, the authors claim that the declines in flight index and climbing activity are due to functional declines in skeletal muscle performance caused by intramuscular lipid infiltration. They argue that the increases in lipid droplet formation within the skeletal muscle on GIO and DIO alters the structural organization and "performance" of the muscle. These results are very interesting and will help move the field forward. However, the main claims within this manuscript need strengthening to meet the standard of a Nat Communication paper.

The authors claim that skeletal muscle physiology and performance is disrupted in the DIO and GIO models, because they observe declines in flight index and climbing performance. Although flight index and climbing performance can be influenced by skeletal muscle structure/function, there are multiple confounding variables that make this experimental setup insufficient in supporting their claims. Firstly, it is possible that the diet or changes in lipid droplet within the skeletal muscle are not causing and functional changes to the skeletal muscle in terms of force generation or myofibrillar structure/organization.

The authors claim that the skeletal muscle of DIO and GIO models induces a disruption in myofibrillar organization, but they only provide longitudinal confocal sections of the skeletal muscle which do not allow for the interpretation of sarcomeric organization/integrity. The mito timer studies are really interesting but it is also known that there is oxidative damage observed in short burst of exercise that can ultimately be beneficial. To make this claim the authors need to perform electron microscopy cross-sections of the skeletal muscle to visualize sarcomeric organization (arrangement of thick and thin filaments). Also examining the mito-UPR maybe useful in this context. The changes in flight index are likely confounded by non-muscle changes induced by high-fat feeding. For instance, it is well established that organisms on HFD demonstrate changes in locomotor activity and feeding behaviors due to changes in the reward pathway of the brain (dopamine, serotonin, etc). Therefore, the authors cannot exclude the possible non-muscle effects on flight index. Another possibility is that the changes in overall weight of the flies with DIO and GIO have profound impacts on their flight performance and climbing activity that are irrespective of changes originating in the skeletal muscle.

The authors claim that the skeletal muscle becomes insulin insensitive with the DIO and GIO models, and this is rescued with TRF. Again, the authors do not directly measure skeletal muscle insulin sensitivity or changes in glucose uptake within the skeletal muscle, but argue their claim by showing increased expression of the NLaz gene that has been associated with insulin resistance. It is very likely that changes in NLaz expression do not, in fact, correlate with changes in insulin sensitivity in their models. Furthermore, the authors could measure the rate of triglyceride turnover and glucose turnover in the muscle to establish their claim.

In summary, the authors need more experimental evidence to support their claim that TRF improves skeletal muscle function by suppressing intramuscular lipid infiltration. Given these shortcomings and the lack of a potential mechanism, the manuscript falls short of the quality of a paper for nature communications.

Reviewer #3 (Remarks to the Author):

Time-restricted feeding mitigates obesity and circadian disruption- induced skeletal muscle decline by suppressing intramuscular lipid infiltration by Villanueva et al.

This is an interesting manuscript that is part of a set of reports exploring the pathophysiological and circadian implications of the time-restricted feeding protocol.

Following the experimental approach of past reports, the authors take advantage of genetic, molecular and physiological tools to accomplish their objectives. Indeed, the aim of the manuscript is to figure out the protective action of TRF in dietary and genetic models of obesity.

Comments to strengthen the manuscript (please check the annexed file to see the punctual points)

1. Results, lines 173-176. Please indicate the time of the day that NLaz was quantified to specify if the sampling was done before, after or during feeding time.

2. Line 193, Mitotimer has been used to study mitochondrial turn over. To explore ROS status within mitochondria, it is more convenient to use Mitosox.

3. Line 206, The use of mitotimer to study mitochondrial stress can be improved if a marker of mitochondrial membrane potential is used instead (Mitotracker, JC-1, etc).

4. Line 262, Fat bodies are equivalent to vertebrate adipose tissue, why not to study adipocytes and not only lipid droplets ??

5. Line 298, It would be interesting to test if genetic/diet conditions of obesity and light/light protocol affect circadian rhythmicity in flying muscles, and then to test if TRF can improve the circadian response in the muscular tissue.

6. Discussion, line 323, please include a reference regarding the presence of inflammation, oxidative stress and tissue dysfunction are associated to hypercaloric feeding in flies.

7. Line 339. Indeed, the fly and climbing tests are suggestive of muscular limitations, but obesity and insulin signaling can also affect neuronal and neuromuscular functions. Please comment.

8. Line 356, to support the statement that FRL is a circadian-protective protocol in the Drosophila IFM, it is necessary to demonstrate daily responses in this tissue and its alteration by hypercaloric feeding.

Minor point.

Jesus Villanueva, in Spanish Jesus is accentuated: Jesús

Reviewer #1: The primary concerns are the following:

1) In general the outcomes assays are very descriptive and do not directly test either muscle function or metabolic sensitivity. This limits the interpretations.

Response: Our changes can be seen under comment 2. We address this concern by providing additional data supporting our muscle function (including muscle specific *knock-down* of *SK2*, Fig. 3) and metabolic data. New data include an alternative method for insulin sensitivity, by directly measuring hemolymph glucose (Figs. 1F and 6C), ATGL-mediated lipid turnover (Fig. 4), EM data for the myofibrillar and mitochondrial dysfunction along with TRF roles in suppressing these abnormalities (Figs. 2 and 7 and SI Fig. 1F), and in group circadian activity (SI Fig. 1G). We have also provided a revised interpretation focused on clarifying our data.

2) The authors can conclude that time of feeding improves indices of flight and climbing and some histological aspects of muscle but there are no true muscle function assays in this study. Any reference to improved muscle function must be softened and acknowledged that the read out is downstream and there could be other factors (i.e. innervation) that might be altered with TRF.

Response: Muscle function measurement by flight test and measurement of climbing performance is widely used as readout for muscle function (over 100 publications, including 10+ publications from our lab). Flight test and climbing performance also have been used for screening-based studies to establish muscle specific function (see Schnorrer et al. *Nature* **464**,287–291, 2010; and Perkins et al. *Scientific Data* **1**, 140002, 2014). Under comment 1, we have provided additional evidence to support our data including ultrastructural evidence for both obesity models and tissue specific RNAi knock-down of genetic-induced obesity mutant *SK2*. Additionally, based upon our EM and confocal data, muscle dysfunction is associated with mitochondrial defects (see ref. Rai et al. *J Cell Sci* **127**, 2014, for mitochondrial specific defect that leads to muscle dysfunction), along with myofibrillar disorganization. However, in agreement with reviewer 1 and other reviewers, we acknowledge that other factors, including innervation, non-muscle function or neuronal function cannot be completely ruled out and have clarified that these factors should be considered. (page # 7, third paragraph and page #15, second paragraph).

3) It is unclear why there are some experiments with n=3 while other experiments have n=30-100 (see the legend for Figure 1)? There are large discrepancies in sample number across experiments and this needs to be cleared up.

Response: We have addressed the discrepancies regarding n number and have added this explanation in the revised manuscript (under the results and figures legends). Briefly, we have used n numbers of over 100 for all the flight and genotoxic assays from 3-5 experiments. We have used n=30-50 for weight/activity assays and for the metabolic/cytological assays we have n=3-9. These differences stem from differing needs of each experiment. While historically, flight and geotaxis assays have had large n numbers, other experiments such as weight and activity assays require less n when considering the statistical significance of results. The lower n number for metabolic and cytological assays represent the time expenditure necessary for carrying out these experiments.

4) Expression of NLaz mRNA is used as a marker of insulin resistance. It is not clear how sensitive this marker is? It is not clear why the authors did not measure glucose or more direct measures of changes in glucose transport/metabolism to more clearly define the metabolic disease aspect of these models.

Response: Overexpression of Apolipoprotein D (a human homolog of NLaz) is also known to induce insulin-resistance and directly influence lipid metabolism (Carmo et al. *Endocrinology and Metabolism* **296**, E802-811, 2009), supporting our data. In addition to NLaz mRNA expression, we have now provided an alternative method for measuring the insulin resistance (via haemolymph glucose measurement). Like our results (Fig. 1F), elevated level of glucose in the hemolymph has been reported in *Drosophila* high-sugar diet models (Na, J. et al. *PLoS genetics* **9**, e1003175, 2013). Both assays lead to the same conclusions (i.e. obesity leads to insulin resistance and TRF enhances insulin sensitivity). NLaz expression can be used as a marker for insulin resistance due to this agreement of results. Moreover, metabolic disease aspects of insulin resistance/sensitivity have been further refined for the several metabolic

diseases including obesity, diabetes, cardiovascular and skeletal muscle diseases and we have cited recently published manuscripts for clarity (see second last paragraph of the introduction, page # 4, especially a sentence about a pilot study on prediabetic men, Ref #41).

4) A minor point is that the use of the Sk2 fly strain for obesity should be referenced in the Introduction. References should be checked throughout the manuscript.

Response: In addition to the second paragraph of the introduction section containing two references about the SK2 mutant, we have further defined the usage of the Sk2 mutant for further clarification in the revised manuscript. We have also checked all the references in the revised manuscript for accuracy.

Reviewer #2

The authors claim that skeletal muscle physiology and performance is disrupted in the diet-induced obesity (DIO) and genetic-induced obesity (GIO) models, because they observe declines in flight index and climbing performance. Although flight index and climbing performance can be influenced by skeletal muscle structure/function, there are multiple confounding variables that make this experimental setup insufficient in supporting their claims. Firstly, it is possible that the diet or changes in lipid droplet within the skeletal muscle are not causing and functional changes to the skeletal muscle in terms of force generation or myofibrillar structure/organization.

Response: As indicated to address reviewer 1's concern, flight test and climbing assays have been widely used as readout for muscle function. We have also performed a muscle specific knock-down of SK2 using a *Fln-Gal4* driver (promoter of *flightin* gene expressed in the flight muscle) in helping support the tissue-specificity of our muscle assays. Most of the parameters, including muscle dysfunction, enhanced intramuscular lipid infiltration and insulin resistance are found in the muscle specific knock-down of SK2, which mimics the same traits of the obese mutant (SK2). Therefore, there is a strong suggestion that mutant SK2-induced dysfunction is muscle autonomous. Furthermore, we have also performed electron microscopy analysis of the IFM to demonstrate diet and genetic obesity-induced myofibrillar/mitochondrial defects at the ultrastructural level. As addressed in the following comment, the major diet and genetic obesity-induced defect is mitochondrial with some sarcomere disorganization, which was significantly suppressed under TRF (Figs. 2 and 7 and SI Fig. 1F). Our cytological data also revealed the subtle myofibrillar disorganization, which could be due to enhanced intramuscular lipid infiltration and insulin resistance. Therefore, muscle dysfunction is associated with mitochondrial defects (see ref. Rai et al. *J Cell Sci* **127**, 2014, for mitochondrial specific defect that leads to muscle dysfunction), along with myofibrillar disorganization.

The authors claim that the skeletal muscle of DIO and GIO models induces a disruption in myofibrillar organization, but they only provide longitudinal confocal sections of the skeletal muscle which do not allow for the interpretation of sarcomeric organization/integrity. The mito timer studies are really interesting but it is also known that there is oxidative damage observed in short burst of exercise that can ultimately be beneficial. To make this claim the authors need

to perform electron microscopy cross-sections of the skeletal muscle to visualize sarcomeric organization (arrangement of thick and thin filaments). Also examining the mito-UPR may be useful in this context.

Response: The point about needing ultrastructural information is very well taken. We have now provided new EM ultrastructural data revealing mitochondrial/myofibrillar defects associated with diet and genetic obesity (Figs. 2 and 7 and SI Fig. 1F). Our results show that both obesity models lead to severe mitochondrial defects along with some sarcomere disorganization, including a significant increase in gaps in the arrangement of thick and thin filaments (see Fig. 2D and E, white arrowheads). Moreover, *SK2* mutants (GIO) showed greater sarcomeric disorganization including misalignment of Z-disk and bent sarcomeres (Fig. 2C and E, asterisks) and mitochondrial defects (Fig. 2C and E arrows) compared to DIO, which also correlates with mitochondrial oxidative stress data from our mito-timer measurements (Fig. 1 J, K). Interestingly, we have shown that the obesity-induced ultrastructural abnormalities we observed were significantly ameliorated under TRF (Fig. 7 and supplementary Fig. 1F). Therefore, our TEM evidence not only supports our functional and cytological evidence, but also demonstrates the impact of TRF in suppressing sarcomere disorganization and mitochondrial abnormalities linked with obesity.

Exercise was not included as a factor in our current study. Therefore, EM data along with mito-timer and mitochondrial abnormalities upon muscle specific knock-down of *SK2* (Fig.3) may be indicative of enhanced oxidative stress/mitochondrial dysfunction in both obesity models. Overall, we have discovered that diet and genetic obesity lead to metabolic disorders including mitochondrial defects and oxidative stress which might lead to disrupting of lipid homeostasis, enhanced intramuscular lipid infiltration and the induction of insulin resistance. We believe that analysis of the mitochondrial unfolded protein response (Mito-UPR) is beyond the scope of this paper and is not going to provide any additional information about mitochondrial health and defects beyond that already shown with mito-Timer and mito-GFP along with the ultrastructural evidence. However, we will be exploring the potential role and effects of mito-UPR as well as ER-UPR in future studies.

The changes in flight index are likely confounded by non-muscle changes induced by high-fat feeding. For instance, it is well established that organisms on HFD demonstrate changes in locomotor activity and feeding behaviors due to changes in the reward pathway of the brain (dopamine, serotonin, etc).

Therefore, the authors cannot exclude the possible non-muscle effects on flight index. Another possibility is that the changes in overall weight of the flies with DIO and GIO have profound impacts on their flight performance and climbing activity that are irrespective of changes originating in the skeletal muscle.

Response: In agreement with the reviewer we cannot entirely rule out non-muscle specific factors associated with DIO which may be responsible for the muscle dysfunction and we have included this possibility in the revised manuscript. However, we have shown that overall group circadian activity of the flies under DIO is not different from the age matched control and GIO model (Supplementary Fig. 1G). High percentage of fat in the diet is responsible for altering pathways of the brain and may affect flight and geotaxis response. However, 5% of the fat may

not be sufficient to affect dopamine and/or serotonin levels, which may explain why overall group circadian activity remained the same. Furthermore, our functional and cytological data show that muscle specific knock-down of *SK2* (Fig. 3) is not different from mutant *SK2* and DIO (Figs. 1E and 5D-G), therefore the flight defect in the DIO suggests likely muscle specificity.

Additionally, the reviewer's concern about DIO and GIO weight gain of flies possibly being responsible for profound impacts on flight performance and climbing activity that are irrespective of changes originating in the skeletal muscle is addressed by our data. Our results have shown that *BMM* mutant flies have statistically higher weights compared to both *SK2* mutant and HFD flies (Fig. 1 B, red asterisk). However, *BMM* mutant flies have enhanced flight performance and climbing activity compared to *SK2* mutant and HFD flies (Fig 1E and 1D, red asterisks). The increase in body weight in the mutant flies does not seem to directly correlate with a decrease in muscle function, rather it is obesity specific. More interestingly, obesity-induced weight gain was reduced nearly to wild-type weight under the TRF regimen (Fig. 6A). In support of our finding, a recent pilot-based study revealed that the TRF regimen suppressed body weight and mitigates metabolic disease risk factors in obese men (Gabel et al. *Nutr Healthy Aging* 4, 345-353 (2018)). Additionally, we showed that group circadian activity parameters (e.g. normal walking and neuronal performance) of the *SK2* mutant and HFD flies is not different from age matched wild-type control (Fig. SI 1G). However, muscle performance of the *SK2* mutant and HFD flies are different (Fig. 1D and E).

The authors claim that the skeletal muscle becomes insulin insensitive with the DIO and GIO models, and this is rescued with TRF. Again, the authors do not directly measure skeletal muscle insulin sensitivity or changes in glucose uptake within the skeletal muscle, but argue their claim by showing increased expression of the NLaz gene that has been associated with insulin resistance. It is very likely that changes in NLaz expression do not, in fact, correlate with changes in insulin sensitivity in their models. Furthermore, the authors could measure the rate of triglyceride turnover and glucose turnover in the muscle to establish their claim.

Response: We thank the reviewer for this suggestion on how to strengthen our report. To address this specific concern, in addition to NLaz mRNA expression, we have added an alternative method for quantifying insulin resistance (Figs. 1F and 6C, hemolymph glucose measurement). This measurement provides a more direct measurement of changes in glucose transport/metabolism and more clearly defines the metabolic disease aspect of our obesity models. Interestingly, both assays lead to the same conclusions (i.e. obesity induced insulin resistance and TRF enhance insulin sensitivity). This confirms that NLaz expression can be used as a marker for insulin resistance in our models. Consistent with our results, overexpression of Apolipoprotein D (a human homolog of NLaz) has been known to induce insulin-resistance and directly influence lipid metabolism (see Carmo et al. *Endocrinology and metabolism* 296, E802-811, 2009).

As per the reviewer's suggestion, we have now also measured adipose triglyceride lipase (ATGL)-mediated lipid turnover (Fig. 4), specifically by blocking/inhibiting AGTL (*BMM* fly homolog). In the presence of mutant *BMM*, mutant *SK2*-induced functional and cytological parameters further deteriorated, as shown by enhanced intramuscular lipid infiltration, near complete disorganization of myofibrils, and severe loss of flight performance (Fig. 4). From this,

we concluded that the major triglyceride lipase ATGL influences mutant *SK2*-induced skeletal muscle, possibly by affecting triglyceride turnover.

In summary, the authors need more experimental evidence to support their claim that TRF improves skeletal muscle function by suppressing intramuscular lipid infiltration. Given these shortcomings and the lack of a potential mechanism, the manuscript falls short of the quality of a paper for nature communications.

Response: We have provided additional data to support our muscle function (including muscle specific *knock-down* of *SK2*, Fig. 3) and metabolic data, including an alternative method for measuring insulin sensitivity (Figs. 1F and 6C): ATGL-mediated lipid turnover (Fig. 4). Additionally, we have provided EM data for the obesity-induced myofibrillar and mitochondrial abnormalities and their attenuation under TRF (Figs. 2 and 7 and SI Fig. 1F). We appreciate the reviewer's comments and suggestions and we believe that we have addressed our original manuscript's shortcomings and have significantly strengthened our manuscript, making it suitable for publication in *Nature Communications*.

Reviewer #3 comments to strengthen the manuscript

1. Results, lines 173-176. Please indicate the time of the day that NLaz was quantified to specify if the sampling was done before, after or during feeding time.

Response: Sampling for the all the NLaz and hemolymph glucose quantification was carried out at zeitgeber time 12 (ZT 12), during day time (9 AM) when light was on in both ALF and TRF flies. Both ALF and TRF flies were transferred to 1.1 % agar vials for 12-hour before sampling as previously described for other metabolic assays (see, Gill et al. *Science* **347**, 2015)

2. Line 193, Mitotimer has been used to study mitochondrial turn over. To explore ROS status within mitochondria, it is more convenient to use Mitosox.

Response: In addition to mito-timer staining which responds to mitochondrial oxidation, we have used more direct evidence using TEM to visualize obesity-induced mitochondrial abnormalities at the ultrastructural level (Figs. 2 and 7 and SI Fig. 1F). We have also measured mitochondrial abnormalities using mito-GFP upon muscle-specific knock-down of *SK2* (Fig. 3 C and D) to provide evidence of a mitochondrial defect. Mitosox fluorescent dye is specific for detection of mitochondrial superoxide; however, since we cannot rule out the involvement of another ROS in obesity-induced mitochondrial defect/oxidation, we think that multiple measures (EM, mito-GFP, and mito-timer) are most appropriate. We greatly appreciated reviewer suggestion and the specific ROS source associated with obesity-induced skeletal muscle dysfunction will be explored in the future.

3. Line 206, The use of mitotimer to study mitochondrial stress can be improved if a marker of mitochondrial membrane potential is used instead (Mitotracker, JC-1, etc).

Response: Our mito-timer data are supported by ultrastructural evidence using EM (Figs. 2 and 7 and SI Fig. 1F), and observation of mitochondrial defects using mito-GFP upon knock-down of SK2 (Fig. 3 C and D).

4. Line 262, Fat bodies are equivalent to vertebrate adipose tissue, why not to study adipocytes and not only lipid droplets??

Response: In Fig. 6 G and H, we stained adipose tissue (abdominal fat bodies) of 3-week old ALF flies which exhibited larger and more irregular shaped lipid droplets compared to age-matched TRF. Moreover, both obesity models showed larger and more irregular lipid droplet shapes compared to age matched control. Therefore, like the intramuscular fat, adipose tissue fat was also found to be mis-regulated in both obesity models, which had normal regulation under TRF.

5. Line 298, It would be interesting to test if genetic/diet conditions of obesity and light/light protocol affect circadian rhythmicity in flying muscles, and then to test if TRF can improve the circadian response in the muscular tissue.

Response: The focus of light/light studies shown in Fig. 8, was to first demonstrate whether the disruption of circadian rhythm directly affected skeletal muscle performance and if this led to cytological abnormalities and secondly, to examine if TRF mitigated these abnormalities. In our previous studies we have shown that light/light-induced disruption of circadian rhythms was associated with cardiac abnormalities in *Drosophila* models but such work did not include skeletal muscle until now (see Melkani et al. *FEBS Letters* **591**, 2017). Our results for this study showed that light-induced disruption of circadian rhythm did negatively correlate with muscle functional performance and cytological integrity (Fig. 8). These abnormalities were alleviated to some extent under TRF, suggesting that TRF can improve circadian response in the skeletal muscle tissue. Additionally, most of the parameters including muscle performance, lipid infiltration and insulin resistance were commonly seen in both obesity models and light-induced disruption of circadian rhythm. We observed these ailments to be attenuated under TRF. With this, we believe that circadian disruption might be a common connection for obesity and light induced muscle dysfunction and may result in amelioration of muscle dysfunction under the TRF regimen. Furthermore, in support of our conclusions, it recently has been shown that TRF is able to ameliorate diet-induced obesity and metabolic syndrome in a mouse model that lacks a circadian clock gene (Chaix et al. *Cell metabolism*, 2018).

6. Discussion, line 323, please include a reference regarding the presence of inflammation, oxidative stress and tissue dysfunction are associated to hypercaloric feeding in flies.

Response: References associated with high fat and high sugar diet in *Drosophila* and their

association with inflammation, oxidative and tissue dysfunction have been included (please see references 11-12, 21-22, 26-27).

7.L7. Line9. Indeed, the fly and climbing tests are suggestive of muscular limitations, but obesity and insulin signaling can also affect neuronal and neuromuscular functions. Please comment.

Response: This issue has been addressed above for the reviewers 1 and 2. We have shown that overall group activity of the flies under DIO is not different from age matched control and GIO model (Supplementary Fig. 1G), suggesting that there is at least no neuronal effect on daily activity. A high percentage of fat in the diet can be responsible for altering pathways of the brain and may affect flight and geotaxis response; however, using only 5% extra fat is not likely sufficient to affect dopamine and/or serotonin related pathways. Therefore, overall activity remains the same. Furthermore, our functional and cytological evidence have shown that muscle specific knock-down of *SK2* (Fig. 3) is not different from mutant *SK2* and DIO (Fig. 1 D, E, G, H and I); therefore, the observed defects in the DIO is likely muscle specific.

8. Line 356, to support the statement that FRL is a circadian-protective protocol in the Drosophila IFM, it is necessary to demonstrate daily responses in this tissue and its alteration by hypercaloric feeding.

Response: As indicated in the last paragraph of the results section, most of the parameters including muscle performance, lipid infiltration, and insulin resistance are common in both obesity models (diet and genetic) and light-induced disruption of circadian rhythm. Due to their attenuation under TRF, we believe that circadian disruption might be a common feature for obesity and light-induced muscle dysfunction and in amelioration of muscle dysfunction under the TRF regimen. Furthermore, in support of our conclusions, it recently has been shown that TRF is able to ameliorate diet-induced obesity and metabolic syndrome in a mouse model that lacks a circadian clock gene (Chaix et al. *Cell metabolism*, 2018).

Minor point.

Jesus Villanueva, in Spanish Jesus is accentuated: Jesús

Response: Thanks for sharing this, we have corrected Jesús E. Villanueva in the manuscript.

Reviewers' comments:

Reviewer #2 (Remarks to the Author):

I have carefully considered the revised version of the manuscript which includes new data on glucose homeostasis and muscle structure using EM. However, the key problem in the study remains the novelty and the lack of potential mechanism. Furthermore, the key issue in the manuscript remains the descriptive nature of the study and the overstatements (regarding insulin sensitivity and muscle function measures). The study falls short in terms of publication for Nature communication and just shows incremental improvements on previous elegant work from the Melkani and Panda groups.

Reviewer #3 (Remarks to the Author):

The authors have provided extensive and appropriate answers to all the comments and suggestions done in the first stage of the reviewing process.

I have no further observations.

Reviewer #4 (Remarks to the Author):

(I) In this manuscript, Villanueva et. al. explore how time-restricted feeding impact skeletal muscle dysfunction associated with obesogenic diets and disruption of normal circadian rhythms (through irregular feeding patterns). The authors' data show that time-restricted feeding can improve metabolic and physiological/functional abnormalities in muscle caused by high-fat diet and genetic-induced metabolic dysfunction or circadian disruption of feeding. Experimental evidence for these conclusions is robust and clear. However, some of the other findings highlighted in the manuscript are not as well supported by experimental evidence, which tempers enthusiasm for the overarching conclusions. These include:

- The diet used to promote intramuscular lipid accumulation (a high fat diet) is artificial to most insects, and thus it can be difficult to interpret metabolic responses to such a diet in these animals.
- The link between a high-fat diet and insulin resistance (especially in the muscle itself) is limited.
- There is no underlying mechanism provided linking a signaling mechanism/genes to both metabolic/physiological disruption of muscle function and the rescue by time-restricted feeding. The data incorporating SK2 loss-of-function animals provides a strong, secondary genetic model to go along with obesogenic diet, but these genetic data cannot be overlaid on the diet data to inform on mechanism (in other words, based on the data provided, there is no evidence that SK2 or ATGL plays a role in diet-induced metabolic dysfunction of the muscle and the associated rescue with time-restricted feeding).
- I don't believe the authors have sufficiently proved that the intramuscular accumulation of lipids, which is rescued by time-restricted feeding, is the true driver of diet/genetic-induced muscle dysfunction as implied in the title. A better understanding of a signaling mechanism (described above) required for these changes would help with this issue.

Highlighted above are my major concerns with the manuscript (in it's revised version). As requested by the editors, I don't want to add an additional, isolated review for the authors, so I have focused my review on the response to Reviewer 1's initial concerns (some of which overlap with mine).

(II) In general, the authors have added a substantial amount of new data to the revised manuscript, which have aided in the interpretation of models and helped support the major conclusions drawn in the study. Based on Reviewer 1's initial comments, there are still a few concerns regarding:

(1) Insulin resistance in the high-fat diet model: – The authors did not provide convincing evidence that these flies are indeed insulin resistant in the first version of the manuscript (measuring the expression of a single gene is not enough). They have since added additional data, hemolymph carbohydrate levels, which further support insulin resistance. However, we still don't know if the muscle itself has reduced insulin signaling in this model (for instance reduced pAKT levels in muscle - which would be important for their findings). I am still not entirely convinced these animals display insulin resistance in muscle. Furthermore, just because NLaz expression correlates with high sugar levels does not mean NLaz can be used as a marker for insulin resistance (as stated in the rebuttal).

(2) The use a heterozygote ATGL/Bmm mutant allele (only in the SK2 genetic model) to highlight a requirement for ATGL-mediated lipolysis/lipid turnover in intramuscular lipid accumulation: - Simply reducing gene dose of ATGL/Bmm might not be truly reflective of its role in this process (and has not been previously used in the literature), and thus there would be better genetic and biochemical strategies to test this that could be utilized in both the diet-and genetic-induced lipid accumulation models.

However, as Reviewer 1's comments were general and did not detail specific assays, the additional data provided would seem to alleviate some of the concerns and justify publication based on the initial review.

Reviewer #4 (comments and responses)

-The diet used to promote intramuscular lipid accumulation (a high fat diet) is artificial to most insects, and thus it can be difficult to interpret metabolic responses to such a diet in these animals.

We agree that the energy dense Diet Induced Obesity models in both vertebrate and insect model organisms is an unusual diet for these animals. However, they also model the unusual energy dense diet that has become common for human consumption. Hence, we and other researchers who use animal models of DIO believe that this artificial diet models the change in human nutrition that has happened over a short period of past few decades in the long history of human existence over thousands of years.

-The link between a high-fat diet and insulin resistance (especially in the muscle itself) is limited.

With additional data including phospho-AKT level, we have addressed this in response to reviewer 4 comments 1 (see below). In brief, we have found that both DIO and GIO models showed significantly elevated level of p-AKT compare to age-matched wild-type control under ALF (Figure 6 g and h). Consistence with our finding, elevated p-AKT has been reported in mouse skeletal muscle and liver tissues under obesogenic challenges (see refs 61). Based on our findings, we concluded that obesity is precursor to insulin resistance, cause p-AKT induction.

-There is no underlying mechanism provided linking a signaling mechanism/genes to both metabolic/physiological disruption of muscle function and the rescue by time-restricted feeding. The data incorporating SK2 loss-of-function animals provides a strong, secondary genetic model to go along with obesogenic diet, but these genetic data cannot be overlaid on the diet data to inform on mechanism (in other words, based on the data provided, there is no evidence that SK2 or ATGL plays a role in diet-induced metabolic dysfunction of the muscle and the associated rescue with time-restricted feeding).

We believe with new data, we have addressed a potential mechanism for TRF-mediated suppression of metabolic/physiological disruption, as indicated below in response to the comments made by reviewers 4 and 2 and in the revised manuscript. Briefly, we have also shown that p-AKT levels were found significantly reduced under TRF in control and obesity

models, compare to their ALF counterparts (Figure 6 g and h). Moreover, we have shown that TRF ameliorate these abnormalities and enhanced muscle performance by enhancing insulin sensitivity, suppressing oxidative stress, intramuscular fat infiltration and impaired p-AKT.

As addressed in response to the 2nd comment made by reviewer 4, we have performed ATGL-mediated lipid turnover and the regulation of muscle dysfunction under diet-induced obesity condition (updated Figure 4). We have also used pharmacological approach to inhibit ATGL (Bmm) function *in vivo* using Bromoenol Lactone (Supplementary Figure 3). Both conditions lead to higher level of intramuscular lipid accumulation, myofibrillar disorganization and skeletal muscle dysfunction. Thus, using, genetic interaction, diet and pharmacological approaches, we have shown ATGL-mediated lipid turnover and the regulation of muscle dysfunction in response to obesogenic challenges.

-I don't believe the authors have sufficiently proved that the intramuscular accumulation of lipids, which is rescued by time-restricted feeding, is the true driver of diet/genetic-induced muscle dysfunction as implied in the title. A better understanding of a signaling mechanism (described above) required for these changes would help with this issue.

We have seen intramuscular lipid accumulation with tissue specific knock-down of the *Sk2*, pharmacological inhibition of ATGL and light-induced circadian disruption. Moreover, we also find TRF ameliorates these abnormalities and enhanced muscle performance and reduced intramuscular fat infiltration. These independent lines of correlations between intramuscular lipid deposit and muscle functions support the idea that tonic increase in IMCL might contribute to compromised muscle function. We agree, more work to be done beyond the scope of this manuscript to test whether causal mechanism underlies this correlation.

Reviewer #4 (additional comments to authors and responses)

(1) Insulin resistance in the high-fat diet model: – The authors did not provide convincing evidence that these flies are indeed insulin resistant in the first version of the manuscript (measuring the expression of a single gene is not enough). They have since added additional data, hemolymph carbohydrate levels, which further support insulin resistance. However, we still don't know if the muscle itself has reduced insulin signaling in this model (for instance reduced pAKT levels in muscle - which would be important for their findings). I am still not entirely

convinced these animals display insulin resistance in muscle. Furthermore, just because NLaz expression correlates with high sugar levels does not mean NLaz can be used as a marker for insulin resistance (as stated in the rebuttal).

Based on the reviewer suggestion, we have measured phospho-AKT (p-AKT) level (one of the main components of signaling pathways involved in metabolic regulation), in DIO and GIO models under ALF and TRF. We have found that both DIO and GIO models showed significantly elevated level of p-AKT compare to age-matched wild-type control under ALF (Figure 6 g and h). Consistent with our finding, elevated p-AKT has been reported in mouse skeletal muscle and liver tissues under obesogenic challenges (see refs 59). We also found p-AKT levels were significantly reduced under TRF in control and obesity models, compare to their ALF counterparts (Figure 6 g and h). This new p-AKT data accompanied by NLaz and hemolymph glucose levels support the interpretation that DIO and GIO resulted in insulin resistance. Based on our findings, we concluded that obesity is precursor to insulin resistance, cause p-AKT induction.

(2) The use a heterozygote ATGL/Bmm mutant allele (only in the SK2 genetic model) to highlight a requirement for ATGL-mediated lipolysis/lipid turnover in intramuscular lipid accumulation: - Simply reducing gene dose of ATGL/Bmm might not be truly reflective of its role in this process (and has not been previously used in the literature), and thus there would be better genetic and biochemical strategies to test this that could be utilized in both the diet-and genetic-induced lipid accumulation models.

We agree that with the reviewer that simply reducing gene dose might be not be true reflective of ATGL-mediated lipid turnover and the regulation of muscle dysfunction under genetic obesity. Based upon reviewer suggestions, we have performed ATGL-mediated lipid turnover and the regulation of muscle dysfunction under diet-induced obesity condition (updated Figure 4). We have found that diet-induced obesity muscle dysfunction was further deteriorated in the presence of *Bmm* mutant (Updated Figure 4a, b). Compared to *Bmm*/+ or HFD-induced muscle function was significantly reduced when HFD was used on *Bmm*/+ background (Fig. 4a, b). These results were consistence, as we have seen in transheterozygotes (*SK2*/+; *Bmm*/+) mutant (genetic obesity mutation). Additionally, under diet-induced obesity condition, aberrant intramuscular lipid infiltration and myofibrillar disorganization was further enhanced, when

carried out in the *Bmm* mutant background (updated Figure 4c, d). Compared to HFD or *Bmm*/+, there were more lipid accumulation and severe myofibrillar disorganization, when HFD was used on the *Bmm*/+ mutant background (updated Figure 4c, d). Overall, these data confirm that inhibition of ATGL contributed additional lipid infiltration and the deterioration of muscle physiology linked with DIO and GIO.

We have also used pharmacological approach to inhibit ATGL (*Bmm*) function *in vivo* using Bromoenol Lactone (BEL). Inhibition of ATGL with BEL resulted in cytological and functional abnormalities, which were more severe in the DIO and GIO compared to age matched control (Supplementary Figure 3). Therefore, using, genetic interaction, diet and pharmacological approaches, we find ATGL-mediated lipid turnover are tightly linked to muscle dysfunction in response to obesogenic challenges.

Reviewer #2

I have carefully considered the revised version of the manuscript which includes new data on glucose homeostasis and muscle structure using EM. However, the key problem in the study remains the novelty and the lack of potential mechanism. Furthermore, the key issue in the manuscript remains the descriptive nature of the study and the overstatements (regarding insulin sensitivity and muscle function measures). The study falls short in terms of publication for Nature communication and just shows incremental improvements on previous elegant work from the Melkani and Panda groups.

We respectfully disagree with the reviewers' overall assessment.

We have addressed the comment about overstatement and have dampened our interpretation in line with the results. However, we believe the manuscript has several objective novelties that was not explicitly stated in the previous submission. Now, these novelties are mentioned in the discussion section. First of all, this is the first study that modeled DIO, GIO and circadian disruption in a *Drosophila* model and directly compared their independent and combinatorial effect on health. Second, we find all three causes of metabolic diseases affect muscle function and structure. Third, we find TRF can attenuate metabolic deterioration in all these three models and improve muscle performance. Specifically, the benefits of TRF on circadian disruption is completely new and will have significant impact on modeling similar studies in pre-clinical animal models or in human clinical studies on shift workers. Given shift workers account for

~20% of the workforce in western countries and shiftwork is often an exclusion criterion for many clinical studies, TRF as a potential intervention to improve health for shift workers is impactful.

Although several papers have shown the impact of DIO, GIO and Circadian disruption on health, most of them (including several from Panda lab) have primarily focused on the liver or adipose tissues, while muscle is the largest metabolic organ. Therefore, this manuscript, even if it is descriptive to some extent, sets an experimental framework for future mechanistic studies.

REVIEWERS' COMMENTS:

Reviewer #2 (Remarks to the Author):

The authors have done a commendable job in responding to the reviewer's comments. The manuscript is now much improved. I agree with the authors though some aspects remain descriptive this work will set the framework for future studies in this area.

Reviewer #4 (Remarks to the Author):

The authors have sufficiently addressed my concerns.